



# Large-eddy simulation of a two-layer boundary-layer cloud system from the Arctic Ocean 2018 expedition

Ines Bulatovic[1], Julien Savre[2], Michael Tjernström[1], Caroline Leck[1] and Annica M. L. Ekman[1]

[1]Department of Meteorology and Bolin Centre for Climate Research, Stockholm University, Stockholm 106 91, Sweden
[2]Meteorological Institute, Fakultät für Physik, Ludwig-Maximilians-Universität Munich, Munich 80333, Germany

*Correspondence to:* Ines Bulatovic (ines.bulatovic@misu.su.se)

**Abstract.** Climate change is particularly noticeable in the Arctic. The most common type of cloud at these latitudes is mixed-phase stratocumulus. These clouds occur frequently and persistently during all seasons and play a critical role in the Arctic energy budget. Previous observations in the central (north of 80° N) Arctic have shown a high occurrence of prolonged periods of a shallow, single-layer mixed-phase stratocumulus at the top of the boundary layer (BL; altitudes ~300 to 400 m). However, recent observations from the summer of 2018 instead showed a prevalence of a two-layer boundary-layer cloud system. Here we use large-eddy simulation to examine the maintenance of one of the cloud systems observed in the summer of 2018 as well as the sensitivity of the cloud layers to different micro- and macro-scale parameters. We find that the model generally reproduces the observed thermodynamic structure well, with two near-neutrally stratified layers in the BL caused by a low cloud (located within the first few hundred meters) capped by a lower temperature inversion, and an upper cloud layer (based around one kilometer or slightly higher) capped by the main temperature inversion of the BL. The investigated cloud structure is persistent unless there are low aerosol number concentrations ($\leq 5$ cm$^{-3}$), which cause the upper cloud layer to dissipate, or high large-scale wind speeds ($\geq 8.5$ m s$^{-1}$), which erode the lower inversion and the related cloud layer. These types of changes in cloud structure lead to a substantial reduction of the net longwave radiation at the surface due to a lower emissivity or higher altitude of the remaining cloud layer. The findings highlight the importance of better understanding and representing aerosol sources and sinks over the central Arctic Ocean. Furthermore, they underline the significance of meteorological parameters, such as the large-scale wind speed, for maintaining the two-layer boundary-layer cloud structure encountered in the lower atmosphere of the central Arctic.

## 1. Introduction

The Arctic warming since pre-industrial times is at least twice as large as the global mean (Holland and Bitz, 2003; Serreze and Barry, 2011; IPCC, 2021). This phenomenon is referred to as Arctic amplification and is accompanied by a drastic decrease in sea-ice extent (Stroeve et al. 2012; Richter-Menge et al., 2018). Climate models generally reproduce the observed Arctic warming, but there is a large spread in the model-projected future near-surface temperature increase (IPCC, 2007; IPCC, 2013; IPCC, 2018; IPCC, 2021). Reasons for the amplified warming are both external forcings, such as heat and moisture transport from lower latitudes (Graversen et al., 2008; Boeke and Taylor, 2016) as well as numerous local and regional feedback processes, such as the surface albedo and lapse-rate feedbacks (Serreze and Barry, 2011; Taylor et al., 2013; Pithan and Mauritsen, 2014; Stjern et al., 2019), and cloud feedbacks (Holland and Bitz, 2003; Taylor et al., 2013).

Arctic low-level, mixed-phase stratocumulus (MPS) clouds play a unique role in the surface energy budget of the region. In the central Arctic (north of 80° N), the longwave radiative effects dominate the annual average cloud radiative forcing at the surface due to the limited amount of solar radiation at these latitudes and the high surface albedo (Intrieri et al., 2002; Shupe and Intrieri, 2004; Sedlar et al., 2011). This leads to a net warming of the surface by the MPS clouds during most of the year. At the end of the summer season, however, MPS can have a net cooling effect on the surface compared to clear-sky conditions due to higher insolation and a reduced surface albedo. The presence of MPS can also influence the timing of the autumn freeze-up period (Intrieri et al., 2002; Shupe and Intrieri, 2004; Tjernström et al., 2014).

Arctic MPS clouds have a complex structure (e.g., Shupe et al., 2006, 2013). They are characterized by a liquid layer present at the cloud top within which ice crystals form and precipitate (Intrieri et al., 2002; Shupe et al.,



2006; Morrison et al., 2012). The radiative properties of the MPS clouds are primarily governed by the cloud liquid phase (Curry and Ebert, 1992; Persson et al., 2017; Dimitrelos el al., 2020), but the liquid amount is in turn strongly dependent on the amount of cloud ice. Ice crystals can rapidly grow through vapor deposition through the Wegener–Bergeron–Findeisen mechanism (Wegener, 1911; Bergeron, 1935; Findeisen, 1938) and also

remove water vapor through precipitation. However, there are several processes involved in maintaining the liquid layer of the MPS clouds. The radiative cooling at the top of the cloud generates turbulent mixing within and below the cloud layer that enhances the growth of cloud droplets and ice crystals. It also increases the entrainment of air from the free troposphere (Tjernström, 2007; Morrison et al., 2012; Sedlar et al., 2012; Shupe et al., 2013). In the central Arctic, the specific humidity often increases across the cloud top (i.e., a humidity inversion) and thus the

entrainment becomes a source of moisture that sustain the liquid layer (Sedlar et al., 2012; Shupe et al., 2013; Tjernström et al., 2012; Solomon et al., 2014; Dimitrelos et al., 2020).

Aerosol particles suspended in the atmosphere also strongly influence the radiative properties of the MPS clouds (Mauritsen et al., 2011; Birch et al., 2012; Bulatovic et al., 2021). The Arctic is generally relatively pristine (Bigg and Leck, 2001; Leck and Svensson, 2015) which leads to a particularly large sensitivity of the cloud to the

changes in the abundance of cloud condensation nuclei (CCN) and ice-nucleating particles (INP; Bigg and Leck, 2001). Several sources are observed to contribute to the aerosol population over the central Arctic Ocean in summer (Heintzenberg et al., 2015; Leck and Svensson, 2015). Previous observations show that low-level clouds and fogs in the central Arctic contain organic material (marine polymer gels) that originate from open leads in the pack ice (Leck and Bigg, 2005; Orellana et al., 2011; Orellana et al., 2021). The organic material may also come

from biological processes occurring upwind near the marginal ice zone (MIZ; Leck and Persson, 1996; Leck et al., 2002; Chang et al., 2011). Plumes of long-range transported pollution are also observed in helicopter profiles over the pack ice (i.e., in the free troposphere (FT) of the central Arctic; Kupiszewski et al., 2013). The observations by Shupe et al. (2013) show that aerosol concentrations often increase across the inversion capping the boundary layer (BL) and modeling study by Igel et al. (2017) indicates that, if the FT aerosols encounter the

cloud tops, they can be entrained into the BL by cloud-induced mixing. However, other studies suggest that the plumes of particles in most cases occur at high altitudes (2 to 3 km) in the FT so that they are rarely mixed down into the BL and the low-level clouds (e.g., Kupiszewski et al., 2013).

Models on different scales are useful tools to understand the complex interactions within the MPS as well as their influence on the surface energy budget. However, due to the complexity and many unknowns of Arctic MPS

clouds, their representation in models is a challenge. For instance, global climate models have difficulty to simulate correct cloud fractions and this results in biases in the surface energy budget and surface temperature (e.g., Karlsson and Svensson, 2011; Sotiropoulou et al., 2016). Numerical weather prediction models also fail to reproduce the observed cloud fraction and surface energy budget in the central Arctic (Birch et al., 2012; Sotiropoulou et al., 2016; Tjernström et al., 2021). Models often cannot reproduce cloud-free conditions when the

aerosol particle (and thus CCN) concentrations are low in the region (Birch et al., 2012; Stevens et al., 2018). Many models have simplified microphysical schemes and e.g., assume constant droplet number or aerosol concentrations (Wesslén et al., 2014; Bulatovic et al., 2019; Young et al., 2021).

Detailed observations of the Arctic climate system are necessary to evaluate numerical models and to obtain a deeper understanding of the Arctic environment. Over the last 25 years, there have been several extensive field

campaigns at latitudes north of 80° N focusing on the ocean, atmosphere, clouds, aerosols and the microbiological life at the ocean surface (e.g., AOE1991; Leck et al., 1996; AOE1996; Leck et al., 2001; SHEBA; Uttal et al. 2002; AOE2001; Leck et al., 2004; Tjernström et al., 2004a, b; ASCOS; Tjernström et al., 2014; MOCCHA AO2018; Leck et al., 2020; MOSAiC, Shupe et al., 2022). Most of these campaigns were characterized by prolonged periods of a shallow, single-layer stratocumulus at the top of the BL (altitudes ~300 to 400m; e.g.,

Tjernström et al., 2012). However, during the MOCCHA AO2018 expedition, there was a high occurrence of multiple cloud layers (Vüllers et al., 2021). Moreover, the thermodynamic structure of the lower atmosphere was characterized by two predominantly near-neutrally stratified layers below the main capping inversion of the BL; one in the lowest few hundred meters and one around one kilometer or slightly higher. The reason for the two-layer BL structure is likely a combination of surface-based turbulent mixing from below and cloud-top buoyancy-

driven mixing from aloft (e.g., Brooks et al., 2017). In synoptic conditions with a relatively deep BL, these two mixing mechanisms become physically separated, creating a decoupled system (e.g., Sotiropoulou et al., 2014). Cloud scenes consisting of more than one single cloud layer involve even more complex interactions than a single-layer, low-level MPS cloud system. For example, the upper cloud layer may seed the lower layer with ice crystals and also impact the mixing of the lower layer by inhibiting cloud-top cooling of the lower cloud. This system may

therefore have a different impact on the surface energy budget compared to a low cloud without an upper cloud layer.

In this article we use observations carried out during the MOCCHA AO2018 expedition and examine a 12 hour long event (18 August at 12:00 UTC to 18 August at 24:00 UTC) with a two-layer boundary-layer cloud structure. We employ a large-eddy simulation (LES) and compare the simulation results with observations and then use the

LES to explore how the cloud layers are sustained. We also examine how different large-scale forcings and





different aerosol and ice crystal number concentrations may impact the cloud properties and their lifetime. Moreover, we analyze the impact of the clouds on the surface radiative fluxes. As such, we design a case study that we envision can be used for further investigation of the specific vertical cloud structure, providing a baseline case for future LES studies.


## 2. Data and methods

### 2.1. MOCCHA AO2018 campaign

The Microbiology-Ocean-Cloud-Coupling in the High Arctic (MOCCHA) Arctic Ocean 2018 (AO2018) expedition took place in August and September 2018 on the Swedish research icebreaker *I/B Oden* (Leck et al., 2020). The expedition started in Longyearbyen, Svalbard on 1 August. After a transit, the icebreaker was moored to an ice floe near the north pole and drifted with it for one month (14 August – 14 September, henceforth referred to as the "ice drift"), while measurements were taken both on the ice and onboard the ship. After a second transit,
I/B *Oden* returned to Longyearbyen on 21 September. A suite of meteorological observations was carried out; see Vüllers et al. (2021) for a detailed description of the meteorological conditions encountered.

The campaign provides a comprehensive dataset on the state of the atmosphere, aerosol and cloud properties, surface water characteristics and sea ice properties as well as detailed information on their coupling. For this study, only a subset of measurements is used to initialize the model and for comparison with the model simulations
(Prytherch, 2019; Prytherch et al., 2019; Karlsson and Zieger, 2020; Prytherch and Tjernström, 2020; Prytherch, 2021; Vüllers et al., 2021). Vaisala RS41 radiosondes were launched daily at 00:00, 06:00, 12:00, and 18:00 UTC. These provided information on the thermodynamic properties of the vertical atmospheric column. Near-surface temperature and relative humidity were measured with an aspirated Rotronic HMP101 sensor. Observations of cloud properties were obtained from different remote sensing instruments. A Metek MIRA-35 Doppler Cloud
radar, a Halo Photonics Streamline Doppler lidar, A Vaisala CL31 lidar ceilometer and an RPG HATPRO microwave radiometer. Together with the soundings these were processed through the Cloudnet algorithm (Illingworth et al., 2007). Downwelling short- and longwave radiation were measured onboard the ship by Eppley PSP and PIR radiometers. Upwelling longwave radiation was calculated from a skin-surface temperature measured with a Heitronics KT15-II infrared temperature sensor, while upwelling shortwave radiation was
calculated using an albedo estimated from 3 hourly surface images. During a shorter period, upwelling radiation was also measured directly on the ice, corroborating these estimates.

Atmospheric aerosol and cloud particles were measured with a whole-air inlet. The inlet was located at approximately 22 m above sea surface height on the container roof at a 45° angle forward. This enabled the inlet to be faced in the forward direction to maximize both the distance from the sea and the ship's superstructure. To
further minimize the risk of sampling pollution from the ship, *I/B Oden* was turned approximately upwind during sampling. A custom-built differential mobility particle sizer (DMPS) with a sample flow 0.36 dm$^3$ min$^{-1}$ was used to measure the aerosol particle number size distribution in the size range 10 to 921 nm at a time resolution of 9 minutes. More technical details can be found in Karlsson et al. (2022) and the data can be accessed through Karlsson and Zieger (2020). The DMPS combined a Vienna-type differential mobility analyzer (inner/outer r=
25/33.5 mm and a length of 280 mm) with a mixing condensation particle counter (MCPC; Model 1720 from Brechtel Manufacturing Inc., USA).

### 2.2. Model

The simulations in this study were performed with the LES MIMICA (MISU MIT Cloud-Aerosol Model) that solves equations for a non-hydrostatic, anelastic atmospheric system (Savre et al., 2014). In the model, a staggered grid system is employed, the so-called C-Arakawa grid. During the advection step, the conservation of momentum and scalars is ensured. Sub-grid scale turbulence is calculated based on the solution of a turbulent kinetic energy equation (Deardorff, 1974). Sensible and latent heat fluxes are calculated based on a fixed near-surface
temperature and transfer coefficients. The model uses a two-moment bulk microphysics scheme to predict mass and number mixing ratios for five hydrometeor types: cloud droplets, raindrops, ice crystals, snow, and graupel (Seifert and Beheng, 2006). Gamma functions are used for the mass distributions for all hydrometeor types. Supersaturation is calculated with a pseudo-analytic method and the integration of condensation/evaporation is modeled following Morrison and Grabowski (2008). A simple power law is used to calculate the hydrometeor
terminal fall speed, based on the diameter of the particle, which determines the aerosol wet deposition. The model includes a two-moment aerosol module with a prescribed number of lognormal aerosol modes (Ekman et al., 2006). Radiation is calculated following a four-stream radiative transfer solver (Fu and Liou, 1993) that includes


six bands for solar radiation and twelve bands for the infrared part of the spectrum. More details about the model setup for this specific study are found in Sect. 3.2.


### 3. Case study

#### 3.1. The observed case

The simulated case is based on observations made on 18 August 2018, from 12:00 to 24:00 UTC, during the ice drift period (see Sect. 2.1). During this event, a weak low-pressure system within the North American sector of the Arctic was moving southward while a weak high-pressure ridge was approaching *I/B Oden* position (Fig. 1, left). Farther south, over Svalbard, a deepening low-pressure system was moving northward towards *I/B Oden*. The sequence of satellite images in Fig. 1 illustrates this progression. Near mid-day on the 18th, *I/B Oden* was
situated under a deep cloud layer. As the low-pressure system moved away, the deeper clouds slowly withdrew from *I/B Oden* while the frontal clouds north of Svalbard progressively moved closer. During most of the second half of 18 August, *I/B* Oden was lying under a solid stratocumulus layer with a fairly low cloud top.

The distribution and properties of the low-level clouds are shown in more detail in Fig. 2. The main cloud base and cloud top heights were analyzed using a combination of three main cloud radar moments aided by lidar
backscatter from the ceilometer. The cloud scene was dominated by two main cloud layers that developed slowly with time. The cloud base of the lower cloud layer was ranging between zero (fog) to ~100 m and the cloud top height was mostly around 200 – 300 m during the case study. After 19:00 UTC, the cloud exhibited greater time variability, first descending slightly around 21.30 UTC but increasing again to ~400 m at 24:00 UTC. The upper cloud layer was initially located between about 0.8 and 1 km and gradually ascended to lie between ~1.2 and ~1.4
km. Intermittent clouds around 400 to 500 m complicated the cloud scene between 16:00 and 21.30 UTC (Fig. 2d); these clouds are not explicitly studied here.

A slow transition towards the steady-state large-scale conditions associated with the developing stratocumulus deck is clearly seen in the observed near-surface variables (Fig. 3). The air temperature at ~20 m above the surface was slightly below freezing through the second half of 18 August (Fig. 3a). The relative humidity was high
throughout but increased from ~95 to 100 % between 12:00 and 15:00 UTC (Fig. 3b). The wind speed decreased from ~6-7 to around 5 m s$^{-1}$ (not shown) and the wind direction turned slightly from ~200 to 170 degrees (not shown). After 15:00 UTC, all these surface variables remained approximately constant except the surface temperature, which changed from slightly below to slightly above freezing at 16:00 UTC (Fig. 3a). However, relative to the freezing/melting point all temperatures were well within the measurement uncertainty.
The radar reflectivity factor depends on the number and size of different hydrometeor types. Thus, the relatively low reflectivity values (low Db$_z$) during the first half of the case study indicate that there was very little precipitation falling out of the upper cloud before ~16:00 UTC (Fig. 2a and 2b). Starting around 16:00 and until 19:00 UTC, precipitation was falling from the upper layer, as evidenced by the cloud radar reflectivity and Doppler fall velocity (the fall velocity is different from zero below the upper cloud layer). However, the precipitation never
reached the lower cloud; it sublimated in the mostly cloud-free air between the cloud layers, which may be the reason for the scattered temporary cloudiness occurring at ~500 m between 16:00 and 17:00 UTC and around 19:00 UTC (Fig. 2d). Fall velocities near the upper cloud base increased and the velocity spread also increased around 19:00 UTC (Fig. 2b and 2c), meaning there was a larger variability in the velocity values. Modest fall velocities combined with a large velocity spread in the lower parts of the upper cloud layer indicate a mixture of
liquid cloud droplets, with a near-zero fall velocity, and larger precipitating ice particles. This is repeated when precipitation reached the upper boundary of the lower cloud layer. At the same time, the upper cloud continued to be dominated by liquid cloud droplets (near-zero terminal velocity and small velocity spread) confined to the upper parts; this layer became shallower as precipitation formation became more active. However, very small amounts of precipitation reached the surface; only a few instances of weak but non-zero precipitation intensity
were registered at the surface (not shown).

The vertical distribution of clouds is consistent with the time-height cross sections of equivalent virtual potential temperature ($\theta_e$) and relative humidity (with respect to ice) derived from the sounding data (Fig. 4). The $\theta_e$-profile clearly displays the two main inversions. The lower inversion was at an almost constant altitude of a few hundred meters, first subsiding and then ascending very slightly. The upper inversion was stronger and started with a base
around 1 km, roughly coinciding with the upper cloud top. Thereafter, it sharpened and ascended to ~1.4 km around 18:00 UTC. Interestingly, below the ascending upper inversion a well-mixed layer (constant $\theta_e$) gradually formed and deepened to become almost 1 km thick at midnight; much deeper than the upper cloud layer. The relative humidity in the lower cloud layer was near ~100 % throughout the whole event (Fig. 4b). The upper cloud initially had a shallow, only slightly supersaturated (w.r.t ice) layer that formed between ~17:00 and 18:00 UTC
(Fig. 4b) and deepened into a substantially supersaturated layer following a cooling by several degrees (Fig. 4a).

The southerly winds also led to the advection of relatively high aerosol number concentrations compared to the average for the whole ice drift (Fig. 5). The median integrated concentrations for the case study were thus 83 and 73 % higher than for the ice drift for the accumulation and Aitken mode, respectively. Furthermore, the accumulation mode number concentration was higher than that for the Aitken mode, opposite to the ratio of the two modes for most of the ice drift period.

In summary, the upper cloud was a dynamically more active cloud than the lower cloud layer. Cloud-top cooling increased as the thicker clouds in the free troposphere withdrew, which enhanced buoyancy overturning, mixing, and entrainment. While entrainment caused the cloud layer to ascend, enhanced cloud overturning deepened the well-mixed layer within and below the cloud. At the same time, the ascent cooled the whole cloud-mixed layer adiabatically and ice supersaturation eventually reached a level when solid precipitation started forming. At first, precipitation did not reach the lower cloud layer; instead, it sublimated in the cloud-free air, possibly causing additional temporary cloud formation indicated in Fig. 2. However, even later, when the precipitation evidently reached the lower cloud layer, barely any precipitation was observed at the surface. The lower cloud layer continued to be dominated by liquid cloud droplets and also tended to thicken. At least until midnight, the frozen precipitation falling from the upper cloud did not cause the lower cloud to dissipate, probably because the lower cloud layer was too warm (the temperature was around the melting point).

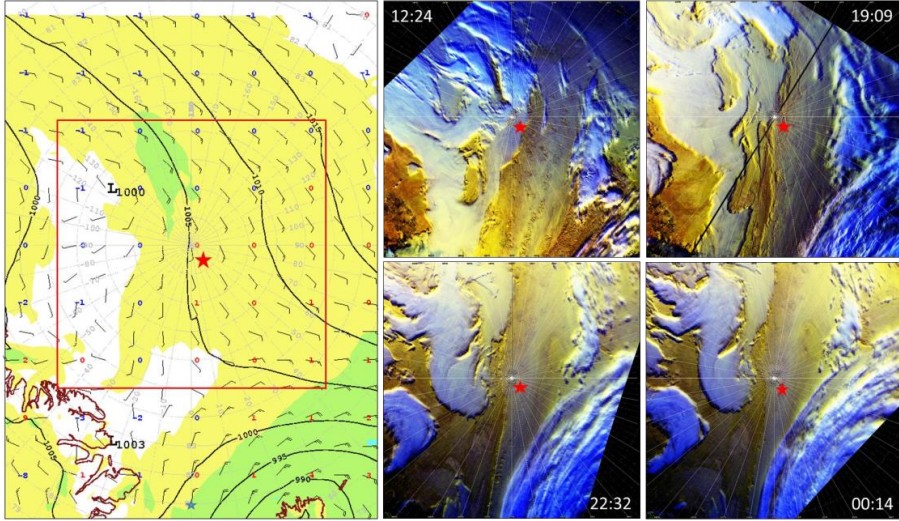

**Figure 1**. (left) surface mean sea-level pressure (solid), wind barbs and wind speed (color) from the ECMWF 12:00 UTC analysis and (right) selection of four satellite false-color images from the NOAA 18 and 19 polar orbiting weather satellites. The red box inside the left figure shows the area of the satellite images, the time for each satellite passage is given in UTC (the first three on 18 August 2018 and the final just after midnight on 19 August 2018) while the red star indicates the position of the icebreaker Oden.



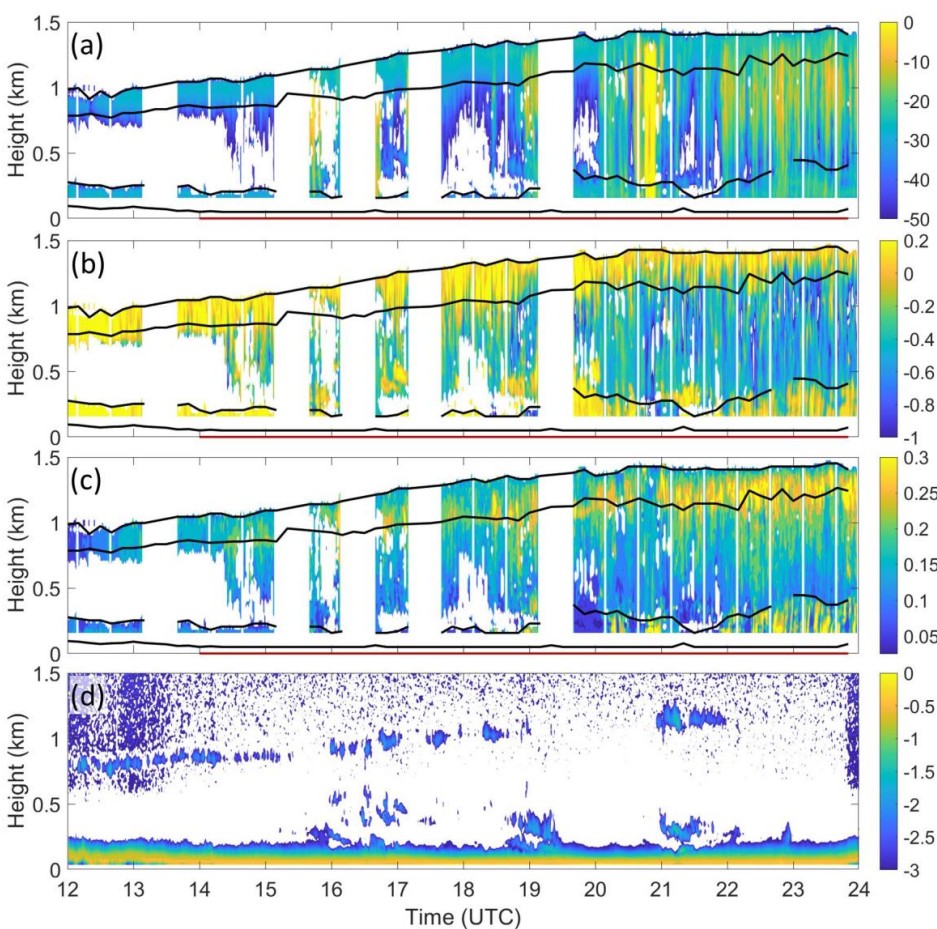

**Figure 2.** Time-height of (a) cloud radar reflectivity (dBze), (b) mean Doppler velocity (m s$^{-1}$), (c) the standard deviation of the Doppler velocity inside each measurement volume (m s$^{-1}$) and (d) lidar backscatter (dBze). The two pairs of lines show the analyzed cloud base and cloud tops using both radar and lidar data, while the red line at zero altitude indicates the time period when the visibility was below 1 km. Note that the height of the cloud base as measured by the lidar depends on the threshold (see main text for more info); when the visibility is below 1 km (the WMO definition for fog) one could say that the cloud base is zero. The broad white portions before 20:00 UTC are when radar data are lacking for technical reasons, while the thinner more regular white bands every half-hour, later on, are from short periods when the radar goes through a scanning pattern and hence is not pointing vertically.





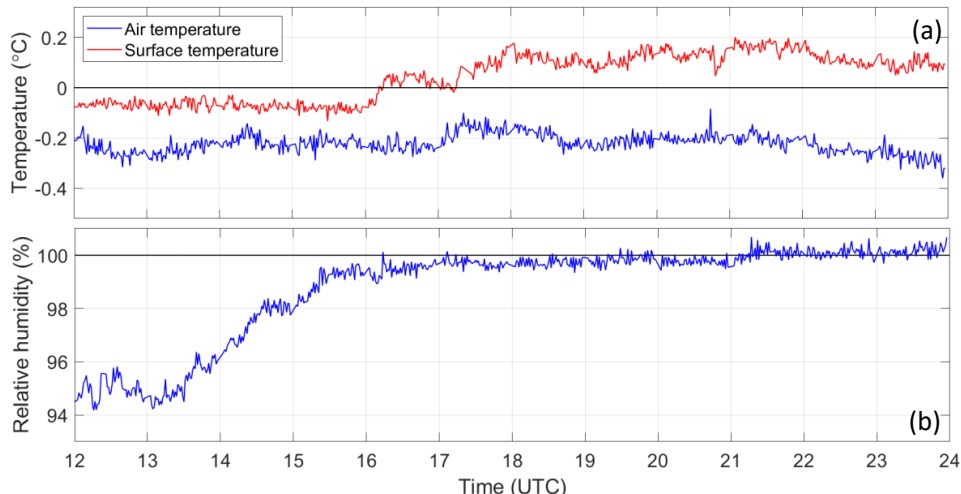

270

**Figure 3.** Time series of selected surface and near-surface variables, showing (a) air (blue) and skin-surface (red) temperatures (°C), (b) relative humidity (%). Observations are from ~20 m above the surface, except the skin-surface temperature which was measured with a downward-looking IR-thermometer located at ~25 m.

275

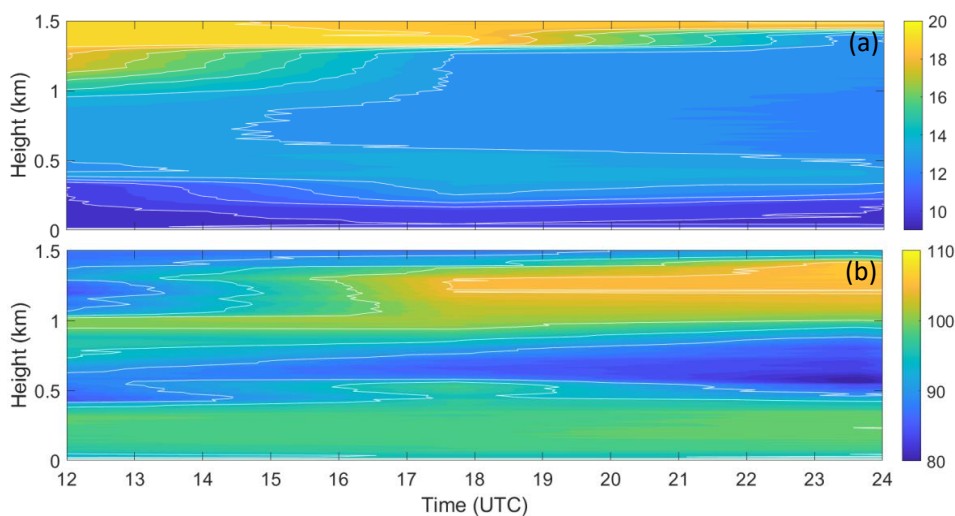

**Figure 4.** Time-height cross-sections of (a) equivalent potential temperature (°C) with white contours every degree, and (b) relative humidity with respect to ice (%) with white contours at 100, 104 and 108 %. Note the coarse temporal resolution based on 6-hourly soundings.

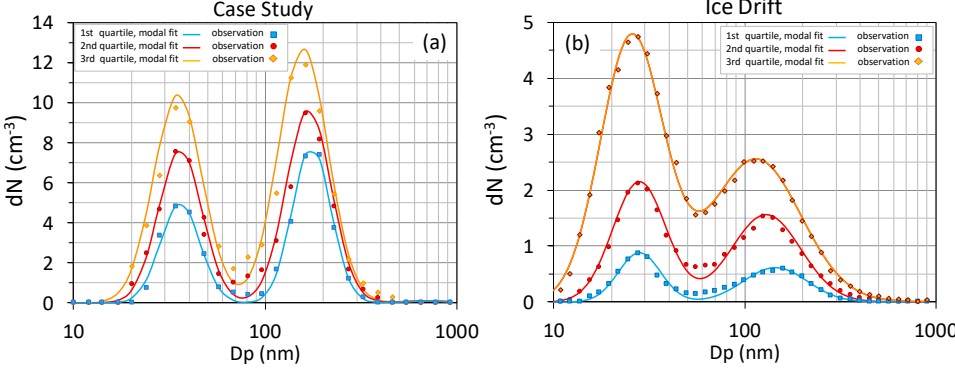

**Figure 5.** Aerosol number size distribution statistics (integrated particle concentration dN vs. modal diameter Dp) shown for (a) the simulated case study (12:00 to 24:00 UTC on 18 August) and (b) the whole ice drift period (14 August - 14 September).

## 3.2. Simulation setup

MIMICA was setup with a horizontal resolution of 62.5 m × 62.5 m covering a domain of 8 km × 8 km. In the control simulation, the vertical grid spacing was variable with a 7.5 m distance in the first 1800 m and then a progressively coarser resolution to the model top at 3 km. The radiosonde data from 18 August 12:00 UTC were used to initialize the pressure, temperature and humidity profiles in the LES. Surface pressure was set to 1005.7 hPa and the surface albedo to 70 %. Both wind components were set to 4 m s$^{-1}$ from the surface to the top of the model domain (corresponding to total scalar wind speed U=5.7 m s$^{-1}$), in rough agreement with observations. The surface temperature was set to 0 °C since the simulated case represents a period with surface melt when any energy surplus leads to the melting of sea ice and snow instead of warming the surface. The large-scale divergence rate



was assumed to be constant within the model domain and prescribed to $1.5 \times 10^{-6}\,\text{s}^{-1}$ to match the observed lifting of the upper cloud layer. The default model time step used in the simulations was 2 s. The total simulation time was 15 h including a 3 h spin-up period to reach the observed state at 12:00 UTC.

The aerosol number size distribution was represented by two log-normal modes (accumulation and Aitken) that were fitted to the observed values (see Fig. 5 and Table A1). In the control simulation, median values

representative of the case study (12:00 to 24:00 UTC on 18 August) were used (Table A1). The observed aerosols were all assumed to be potential CCN and the activation followed the κ-Köhler theory (Petters and Kreidenweis, 2007). The parameter kappa, κ, (Petters and Kreidenweis, 2007), was set to 0.30 and 0.46 to describe the hygroscopic properties of the accumulation and Aitken mode particles, respectively (values were calculated based on the MOCCHA AO2018 observations, pers. comm. with Mike Lawler). Regeneration of aerosol particles upon

droplet and ice crystal evaporation was accounted for, and all regenerated particles were distributed to the accumulation mode (diameters ~80 – 500 nm; Covert et al., 1996). In this study, aerosols were not acting as ice nuclei; ice formed if there was supercooled water present and the ice crystal number concentration was relaxed towards a constant value (Ovchinnikov et al., 2011, 2014). The ice crystal number concentration (ICNC) was set to $0.01\,\text{L}^{-1}$ in rough agreement with INP concentration measurements (Porter et al., 2022). The liquid water content

derived from the thermodynamic profiles, assuming adiabatic conditions, was used for the model initialization; hence, both cloud layers were present at the beginning of the simulation.

### 3.2.1 Sensitivity simulations


In addition to the control simulation, we performed a set of sensitivity experiments (Table 1) to explore how the simulated cloud layers depend on changes in different dynamical and microphysical parameters. To test the sensitivity to the dynamical large-scale forcing, we carried out a set of simulations with different values of the large-scale divergence rate. Since the divergence rate is a parameter that cannot be measured, we used values to

cover a range around the default estimate. Furthermore, we performed a simulation set with different values of the wind speed, with a decrease of ~40 % and an increase of 50 % relative to the control simulation. To test the sensitivity of the simulated cloud structure to the prescribed aerosol number size distribution, we first used the median values for the whole ice-drift period (14 August – 14 September) then the 25th and 75th percentiles of the case study period and finally the 25th and 75th percentiles for the whole ice drift. To investigate the role of the

ICNC, we also performed a set of simulation experiments with different ICNC to represent possible scenarios with ice-free and ice-rich clouds over the central Arctic.

| parameter | value | simulation name |
| --- | --- | --- |
| large-scale divergence ($\text{s}^{-1}$) | $1\times10^{-6}$, **$1.5\times10^{-6}$**, $2\times10^{-6}$, $2.5\times10^{-6}$ | div_1, control, div_2, div_2.5 |
| scalar wind speed ($\text{m s}^{-1}$) | 3.5, **5.7**, 8.5 | wind_3.5, control, wind_8.5 |
| aerosol number size distribution (modal fit) | 25th, **50th**, 75th percentile of the case study, 25th, 50th, 75th percentile of the ice drift | aero_cs_low, control, aero_cs_high, aero_id_low, aero_id_median, aero_id_high |
| ICNC ($\text{L}^{-1}$) | 0, **0.01**, 0.1, 1 | ice_0, control, ice_0.1, ice_1 |

**Table 1.** Simulation summary. Bold letters are used for the values of the control simulation. The sensitivity simulations are named using a combination of the parameter name and the parameter value used in the sensitivity experiment (e.g., aero_cs(id)_low stands for sensitivity to aerosol number concentration taken from the case study (ice drift), lower quartile).




## 4. Results

### 4.1. Control simulation – general features and comparison against observations

MIMICA simulates the general features of both the observed temperature inversions well (Fig. 6a). However, for all three sounding times, the altitude of the inversion capping the lower cloud layer is slightly higher in the model compared to the observations. The free troposphere is colder in the LES than in the observations around midnight (24:00 UTC profile). MIMICA also captures the observed static stability of the BL throughout the simulated period, with a surface-based mixed layer (SML) capped by a lower temperature inversion and an upper mixed layer that is decoupled from the SML. The layer in between the simulated lower and upper inversion is near-neutrally stratified, which agrees well with the observed stratification at these altitudes. The observed moisture profile exhibits larger variability with height than the simulated for all three sounding times (Fig. 6b). The layer between 500 and 1000 m is drier in the observations than in the LES for the second half of the simulated period. Moreover, the observed moisture inversions associated with the lower and upper clouds are slightly stronger in the observations than in the model. The surface sensible and latent heat flux were both small during the case study (Vüllers et al., 2021). Hence, the discrepancy in temperature and moisture between the model and observations is likely due to that large-scale advection is not explicitly considered in the LES. The simulated relative humidity with respect to water and ice, respectively, show that both cloud layers are saturated with respect to water and supersaturated with respect to ice, which is typical for mixed-phased clouds where ice crystals grow at the expense of supercooled water droplets (Fig. A1a and A1b). The simulated radiative heating rates are generally positive within the cloud layers but they are negative (i.e., radiative cooling) in a thin layer at the top of the upper cloud (Fig. A1c). The cooling is also the source of the buoyancy-induced turbulence in the BL (Fig. A1d). The average buoyancy is negative in both cloud layers (cf. Fig. A1d), which means that the cloud motions are dominated by intense and narrow downdrafts compensated by wide but relatively weak updrafts.

The simulated liquid water path (LWP) compares well with the estimate obtained from the microwave radiometer measurements (Fig. 7a). The modeled value is close to the observed range until ~22 h; after that it is up to ~22 % lower than the mean of the retrieved LWP. This difference can be attributed to a weaker lower temperature inversion and a shallower lower cloud layer produced by the model towards the end of the simulation (Fig. 6a and 8a). The temporal variability in the retrieved LWP after ~20:00 UTC, with a decrease followed by a rapid increase, may also be associated with changes in the depth of the lower cloud layer which first thins slowly and then thickens (Fig. 2a) – a feature that is not captured by the model. The average ice water path (IWP) derived from the Cloudnet ice water content profiles shows a similar increase with time as the simulated IWP, although the latter increases faster from ~17:00 to ~21:00 UTC and is consequently larger than observed during the second half of the simulated period (Fig. 7b).

Figure 8a shows that the model simulates two cloud layers, which are located at similar altitudes as the clouds observed by the radar (Fig. 2). The lower cloud base is located between 0 and 100 m and there is cloud droplet water ($q_c$) present up to ~400 m. However, the time variability of the lower cloud top is not well captured by the model. MIMICA simulates a more stationary, slowly decaying, cloud layer while observations indicate a thickening lower cloud. In the model, the upper cloud layer is initially located between ~0.8 and ~1.1 km. The cloud top then gradually ascends with time and reaches ~1.4 km towards the end of the simulation. Most of the liquid water is located at the cloud top, in agreement with observations (Fig. 2a and 2b). The maximum cloud droplet number concentration is ~40 cm$^{-3}$ and ~60 cm$^{-3}$ in the lower and upper cloud layer, respectively (Fig. 8b). The mass and number concentrations of rain, ice crystals and snow are low during the beginning of the simulation and reach a maximum below the upper cloud layer during the second half of the simulated period (Fig. 8c…h). Raindrops fall from the upper layer but most droplets evaporate before they reach the lower cloud. After six hours of simulation, graupel starts forming in significant amounts. The graupel falls out from the upper cloud layer and is the dominant precipitating particle type that reaches the surface (Fig. 8i and 8j).





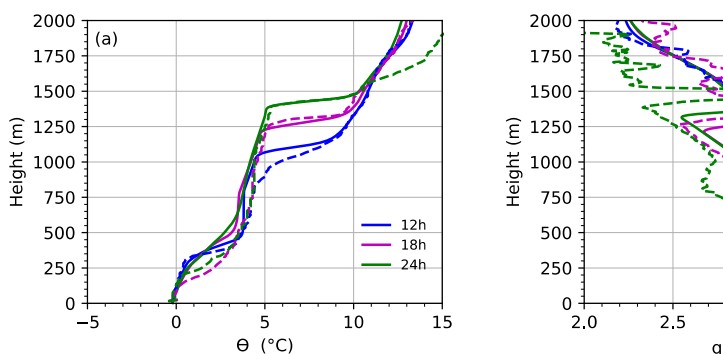

**Figure 6.** Simulated (solid) and observed (dashed) vertical profiles of: (a) potential temperature and (b) water vapor mixing ratio. Note that simulated profiles are horizontal averages over the model domain.

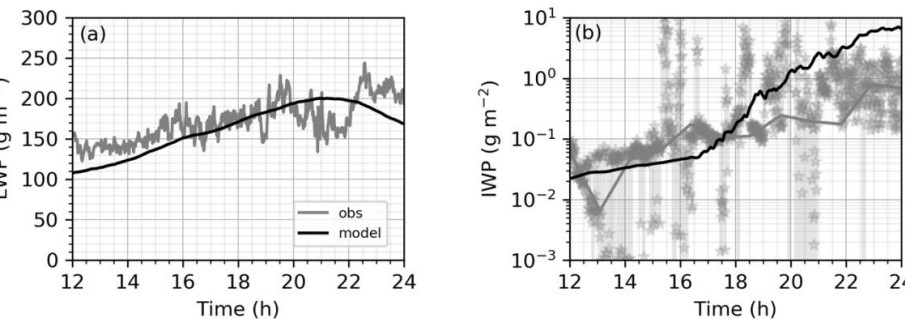

**Figure 7.** Model-domain average simulated and retrieved time series of: (a) LWP and (b) IWP. Raw IWP is shown with a star symbol while the grey line represents the IWP 1h average. The first 3 hours are excluded as they are considered to be a spin-up period.



**Figure 8.** Simulated model-domain average mass (q) and number mixing ratios (N) for the control run. Two prognostic variables are calculated by the model for five different hydrometeor types: (a,b) cloud droplet water (all cloud droplet water with mixing ratio qc > 0.001 g kg$^{-1}$) ; (c,d) rain drops; (e,f) ice crystals; (g,h) snow and (I,j) graupel. The two pairs of lines show the analyzed cloud base and cloud tops using both radar and lidar data. The first 3 hours are excluded as they are considered to be a spin-up period.





### 4.2. Sensitivity simulations

### 4.2.1. Sensitivity to the aerosol size distribution

The relatively high aerosol number concentrations in the case study (Fig. 5) lead to a higher cloud droplet mixing ratio compared to the simulation where the median values for the ice drift are used (control versus aero_id_median; Fig. 9c and 9d). In general, the difference in cloud droplet mixing ratio is largest at the top of the upper cloud layer where most of the cloud droplets are present. Similarly, the simulation aero_cs_low (aero_cs_high) produces less (more) cloud droplet water than the control run, as there are less (more) aerosols available that can serve as CCN (Fig. 9, left column). All experiments using aerosol number concentration data corresponding to the whole ice drift period (aero_id_low, aero_id_median, aero_id_high) produce less cloud droplet water than the control simulation (Fig. 9, right column). The upper cloud top is also consistently lower than in the control simulation, which is due to lower radiative cooling, lower buoyancy production, lower turbulent kinetic energy and lower entrainment rates. Furthermore, the latent heating is different among all the sensitivity experiments due to different condensational rates at the upper cloud layer base. This changes the stability in the BL and also has an impact on the cloud top heights (not shown). In the aero_id_low experiment, the upper cloud layer even dissipates while the lower cloud persists (Fig. 9b).

Among the experiments aero_cs_low, aero_cs_median and aero_cs_high, the most pronounced difference in LWP occurs during the last 6 hours of simulation (Fig. 10a). The experiments aero_id_low, aero_id_median and aero_id_high have substantially lower LWP than the control simulation already at the start of the simulation and the values then stay approximately constant with time. The IWP is generally inversely proportional to the LWP (Fig. 10b). This result may seem counterintuitive as one would expect the IWP to increase as a function of LWP if the ICNC concentration is constant. The reason is that the collection of cloud droplets by ice becomes more efficient in experiments with less LWP (i.e., when the droplets are larger; not shown).

Figure 10c show that the optically thinner clouds with less cloud water transmit more downwelling shortwave radiation to the surface. The control simulation is in fair agreement with the observations with a net surface shortwave flux of approximately 22 W m$^{-2}$. Considerably larger net shortwave fluxes are produced in the aero_id_low, aero_id_median and aero_id_high experiments, up to 48 W m$^{-2}$ (aero_id_low). Most experiments simulate a net surface longwave radiative flux of ~-4 W m$^{-2}$, which is close to the observed estimate (Fig. 10d); this is however controlled by the lower cloud layer as long as it is optically thick. It is only in the simulations where the LWP drops to ~50 g m$^{-2}$ or below (aero_id_median and aero_id_low) that the net longwave radiation drops substantially, by more than 40 W m$^{-2}$ in the aero_id_low experiment. Note, however, that the surface temperature in these simulations is fixed at 0 °C which impedes any compensating drop in surface temperature as the surface energy budget changes.





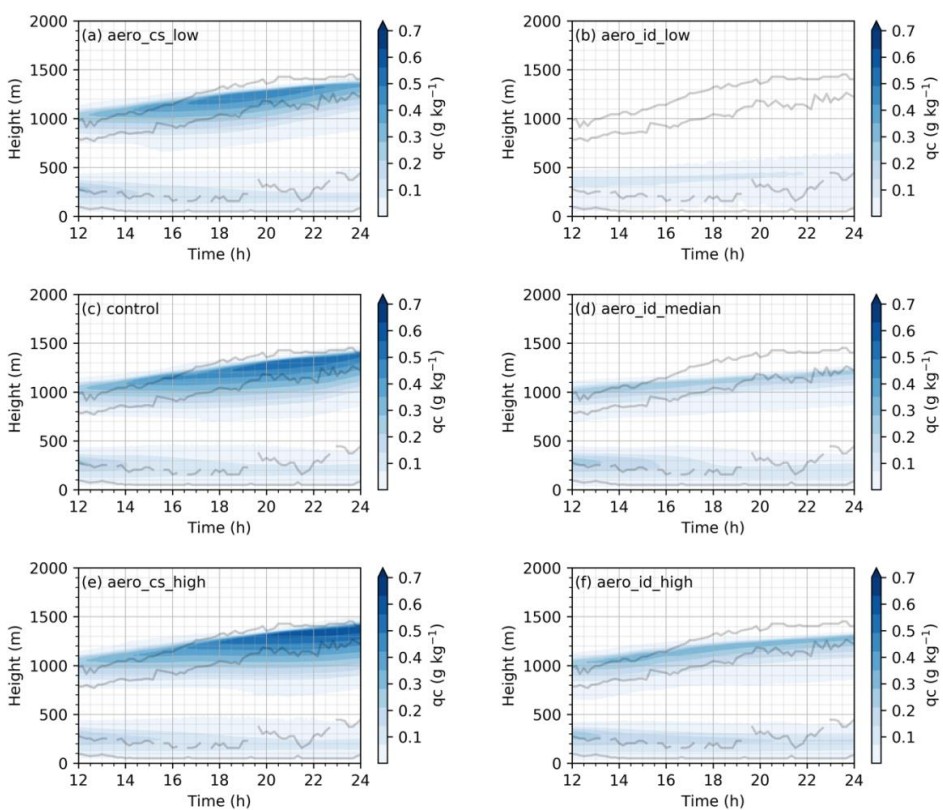

**Figure 9.** Model-domain average cloud droplet mixing ratio from: (left) aero_cs_low, control, aero_cs_high simulations, and (right) aero_id_low, aero_id_median, aero_id_high simulations (see Table 1 for simulation names). The two pairs of lines show the analyzed cloud base and cloud tops using both radar and lidar data. The first 3 hours are excluded as they are considered to be a spin-up period.

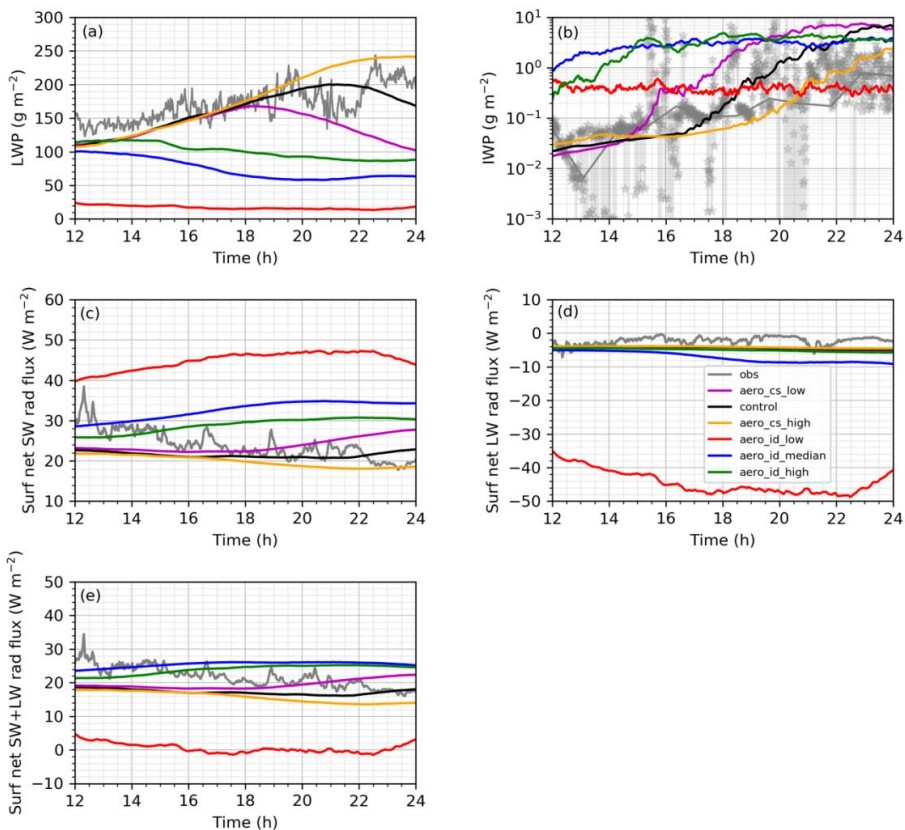


**Figure 10.** Model-domain averaged and retrieved time series of: (a) LWP, (b) IWP, (c) surface net shortwave (SW) radiative flux, (d) surface net longwave (LW) radiative flux and (e) surface net SW+LW radiative flux. The results from different sensitivity tests are in different colors (see the figure legend). The retrieved values from observations (obs) are in grey color. Raw IWP is shown with a star symbol while the grey line represents the IWP 1h average. The first 3 hours are excluded as they are considered to be a spin-up period.


### 4.2.2. Sensitivity to ice crystal number concentration

Sensitivity tests with a higher ICNC produce less liquid water and optically thinner clouds than the control simulation (Fig. 11). However, cloud thinning is not as pronounced as when the aerosol number concentration was reduced (Sect. 4.2.1) and all four sensitivity experiments produce cloud tops at a similar altitude. The largest change in cloud droplet water is obtained in the simulation ice_1. However, even in this case, both cloud layers are sustained throughout the whole simulation period. This amount of ice is thus not enough to glaciate the cloud.

Even though it results in a substantial LWP reduction (Fig. 12a), the changes in the surface radiative fluxes are small. The change in the short- and longwave radiation is less than ~4 W m$^{-2}$ and ~2 W m$^{-2}$, respectively, and occurs only after ~4 hours of simulation (Fig. 12c and 12d).


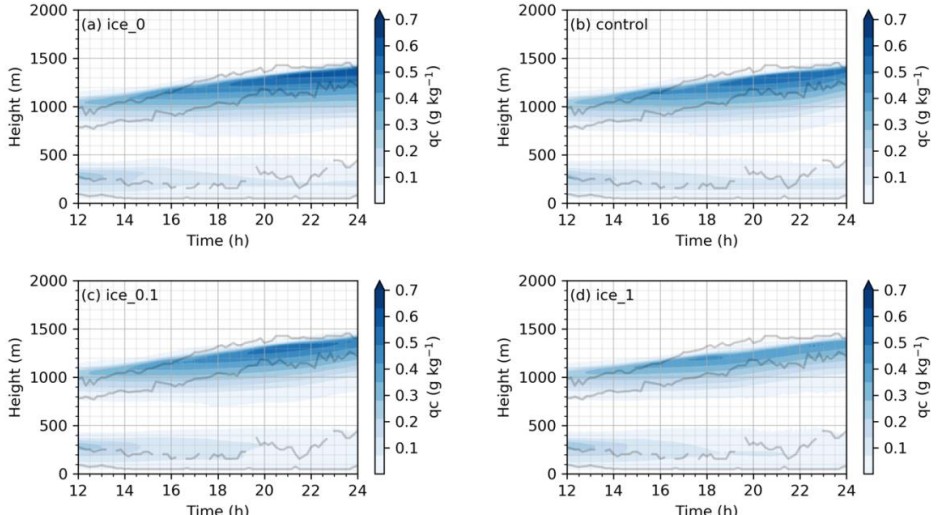


**Figure 11.** Model-domain average cloud droplet mixing ratio simulated with a: (a) no ice (ice_0); (b) the ice crystal number concentration =0.01 L⁻¹ (control simulation); (c) the ice crystal number concentration =0.1 L⁻¹ (ice_0.1); (d) the ice crystal number concentration =1 L⁻¹ (ice_1). The two pairs of lines show the analyzed cloud base and cloud tops using both radar and lidar data. The first 3 hours are excluded as they are considered to be a spin-up period.






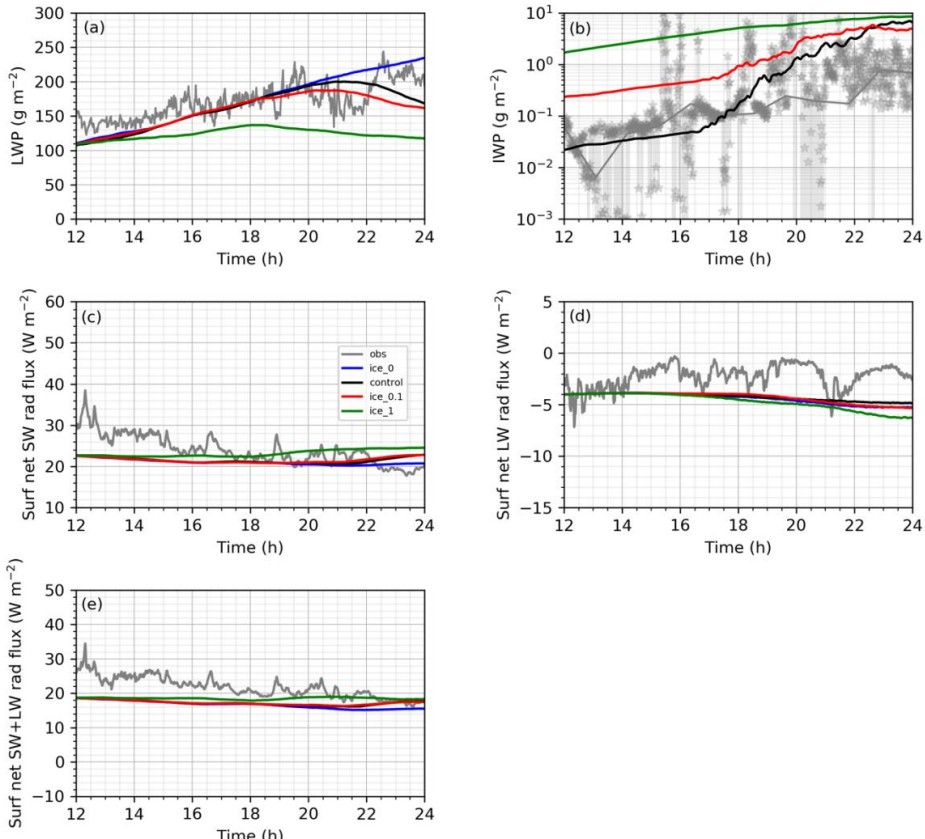


**Figure 12.** Model-domain average simulated and retrieved time series of: (a) LWP, (b) IWP, (c) surface net shortwave (SW) radiative flux, (d) surface net longwave (LW) radiative flux and (e) surface net SW+LW radiative flux. The results from different sensitivity tests are in different colors (see the figure legend). The retrieved values from observations (obs) are in grey color. Raw IWP is shown with a star symbol while the grey line represents the IWP 1h average. The first 3 hours are excluded as they are considered to be a spin-up period.


### 4.2.3. Sensitivity to large-scale divergence and wind speed


With higher large-scale divergence, the turbulent mixing is slightly suppressed (not shown) leading to a shallower upper cloud with a lower cloud top (Fig. 13c and 13d). For the div_1 experiment, the amount of cloud droplet water is higher than in the control simulation (Fig. 13a and 13b). However, the differences are relatively small and do not lead to substantial changes in the LWP, IWP or surface radiative budget (Fig. 14).

A change in wind speed impacts both cloud layers, both in terms of cloud thickness and the amount of cloud droplet water (Fig. 15). With stronger winds (i.e., wind_8.5 simulation), the lower cloud layer is not sustained throughout the simulation; it starts to dissipate after ~4 hours of simulation (Fig. 15c) when the lower temperature inversion erodes (not shown). The upper cloud layer is generally located at a higher (lower) altitude when the wind speed is stronger (weaker). This is associated with a weaker (sharper) temperature inversion capping the

upper cloud (not shown). However, the amount of cloud droplet water at the top of the upper cloud layer is similar in all three simulations (~0.5 to 0.7 g kg⁻¹). The simulated LWP and IWP generally increase with decreasing wind speed (Fig. 14a and 14b). In the wind_8.5 experiment, the total amount of water increases with time but the net





longwave radiative flux at the surface becomes more negative with time (Fig. 14d). This means that the absence of a lower cloud layer has a larger impact on the net surface longwave radiative flux than the total amount of water present in the vertical column above the surface. In other words, the lower cloud layer has a larger impact on the surface longwave radiative flux than the upper cloud because of its higher temperature as long as its total liquid water is large enough to make the cloud radiate as a black body. The control and wind_3.5 simulations produce similar net longwave radiative fluxes at the surface, and they are also close to the observations. The surface net shortwave radiative flux is between ~20 and ~25 W m$^{-2}$ in all three simulations (Fig. 14c). Most shortwave radiation transmitted to the surface is obtained in the experiment with the strongest wind speed (wind_8.5).

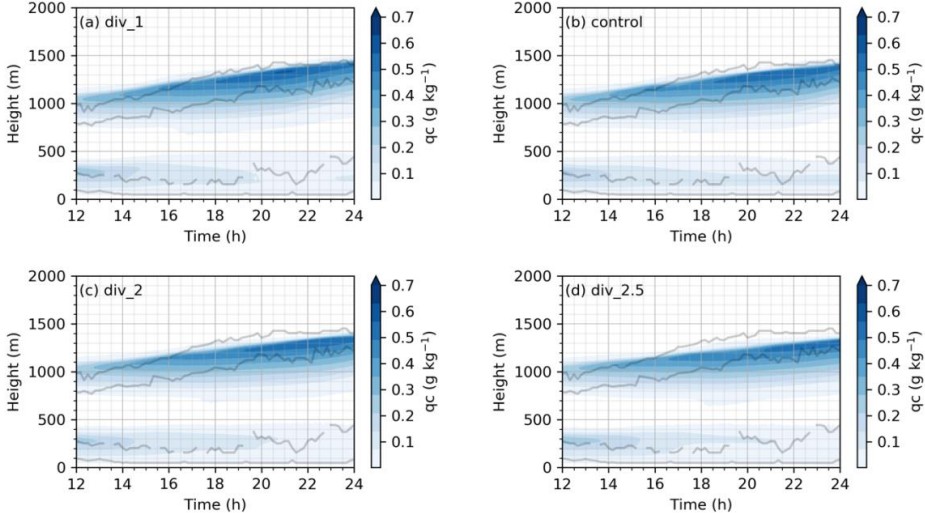

**Figure 13.** Model-domain average cloud droplet mixing ratio simulated with a: (a) divergence = 1 x 10$^{-6}$ s$^{-1}$ (div_1); (b) divergence =1.5 x 10$^{-6}$ s$^{-1}$ (control simulation); (c) divergence =2 x 10$^{-6}$ s$^{-1}$ (div_2) and (d) divergence =2.5 x 10$^{-6}$ s$^{-1}$ (div_2.5). The two pairs of lines show the analyzed cloud base and cloud tops using both radar and lidar data. The first 3 hours are excluded as they are considered to be a spin-up period.



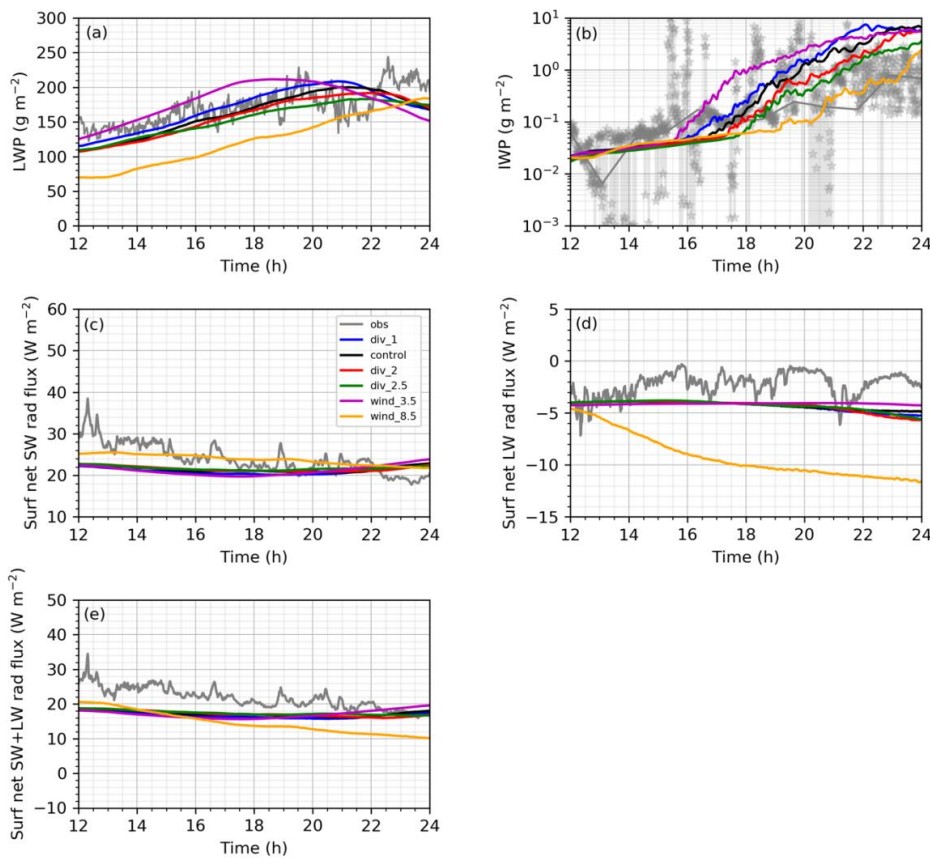


**Figure 14.** Model-domain average simulated and retrieved time series of: (a) LWP, (b) IWP, (c) surface net shortwave (SW) radiative flux, (d) surface net longwave (LW) radiative flux and (e) surface net SW+LW radiative flux. The results from different sensitivity tests are in different colors (see the figure legend). The retrieved values from observations (obs) are in grey color. Raw IWP is shown with a star symbol while the grey line represents the IWP 1h average. The first 3 hours are excluded as they are considered to be a spin-up period.






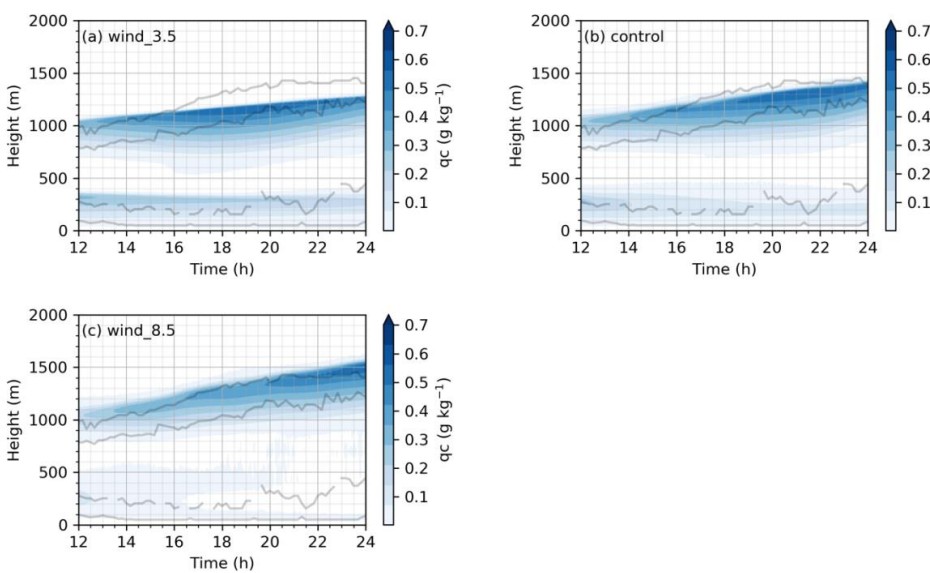


**Figure 15.** Model-domain average cloud droplet mixing ratio simulated with a: (a) U=3.5 m s$^{-1}$ (wind_3.5); (b) U=5.7 m s$^{-1}$ (control simulation) and (c) U=8.5m s$^{-1}$ (wind_8.5). The two pairs of lines show the analyzed cloud base and cloud tops using both radar and lidar data. The first 3 hours are excluded as they are considered to be a spin-up period.


**5. Summary and conclusions**

We have used large-eddy simulation to examine the processes that maintain and influence a two-layer mixed-phase boundary-layer cloud structure, frequently observed in the central Arctic during the MOCCHA AO2018 expedition (Leck et al., 2020). The simulations were conducted for a twelve-hour long case study (12:00 to 24:00 UTC on 18 August) when conditions were relatively stationary, with moderate winds from south/southeast and relatively high number concentrations of aerosols. The case was characterized by one static cloud layer below 500 m and an upper ascending cloud layer with a cloud top rising from ~1 km to slightly below 1.5 km. The Aitken
and accumulation mode median values were around 7 cm$^{-3}$ and 9 cm$^{-3}$, respectively, which was about 73 and 83 % higher compared to the median values for the whole ice drift period.
        The observed cloud properties and the BL vertical structure were in general well-reproduced by the model when the simulations were initialized with thermodynamic profiles and winds based on soundings and aerosol concentrations and properties based on surface observations. Both the model and the observations showed that
the lower cloud was located within a surface-based mixed layer that was capped by an inversion, which remained at about the same height during the case study. The model and observations also agreed that there was a near-neutral layer above the lower inversion, which was generated by the upper cloud layer that in turn was capped by an upper inversion. Small differences between the simulated and observed humidity profiles were likely due to the fact that large-scale horizontal advection was not explicitly considered in the model.
The simulated LWP compared well with the retrieved value during the major part of the simulation period. The IWP in the model increased with time at a rate similar to the Cloudnet-estimated IWP. However, the simulated IWP increased faster from ~17:00 to ~21:00 UTC and was thus larger than the observations during the second half of the simulation. The bulk of the modeled cloud droplet water was located at the top of the upper cloud layer. Throughout the whole case study, the model produced surface net shortwave and longwave fluxes that were in
good agreement with observations.
        We performed sensitivity simulations with the model, where we changed the aerosol and ice crystal number concentrations, the large-scale divergence, and the wind speed. The main conclusions are:





- It was mainly the solar part of the spectrum that was affected when the aerosol number concentration was changed within the range of the observed variability of the case study. A lower aerosol concentration resulted in a slightly lower LWP, for both the upper and lower cloud, and a net warming of the surface (for ~4 W m$^{-2}$ at the end of the simulation period).
- When Aitken and accumulation mode aerosol number concentrations in. being representative of the whole ice drift period (instead of the case study) were used, then the total LWP decreased substantially (up to 150 g m$^{-2}$ at the end of the simulation period). If the model was initialized with aerosol number concentrations representative of the lowest observed percentile (=5 cm$^{-3}$), then the upper cloud layer dissipated and the LWP of the lower cloud decreased to about 20 g m$^{-2}$. This means that the lower layer became partially transparent to radiation so that the longwave radiative effect dominated instead of the shortwave effect, i.e., a lower aerosol concentration resulted in a cooling of the surface.
- Changes in wind speed affected both the upper and lower cloud layers. Increased wind speeds (from 5.7 to 8.5 m s$^{-1}$) resulted in a slightly elevated upper cloud layer and an erosion of the lower inversion so that the lower cloud dissipated after ~4 h of simulation. The lower LWP and IWP in this experiment caused a drop in the surface net longwave radiation by 6 to 7 W m$^{-2}$ at the end of the simulation. There was no substantial effect on the surface net shortwave radiation.
- A change in the ice crystal number concentration from 0 to 1 L$^{-1}$ had no substantial effect on the net radiative flux at the surface; even in the experiment with ice crystal number concentration of 1 L$^{-1}$, the LWP was above 100 g m$^{-2}$. Increased ice crystal number concentrations initially caused an order of magnitude increase in the IWP, but these increases became smaller with time and essentially vanished towards the end of the simulation.
- Changes in the large-scale divergence had no substantial effect on the surface radiative budget. Changes in divergence altered the vertical location of the upper cloud layer while the effects on the lower cloud were small. Lower divergence generally increased cloud droplet water and *vice versa* but the effects on LWP and IWP were not large enough to have any substantial effect on the surface radiation.

In summary, changes that substantially altered the LWP were found to affect the surface net shortwave radiation, but mostly not the surface net longwave radiation. A noticeable effect on the longwave radiation was only obtained in the experiments where the lower cloud layer became optically thin or completely dissipated (i.e., the simulations with the lowest aerosol number concentrations and the highest wind speed, respectively). A reduction in the aerosol number concentration from the highest to the lowest observed values therefore first resulted in an indirect surface warming effect and then a surface cooling effect, in line with the results by Mauritsen et al. (2011).

The absence of the upper cloud layer in the simulation with the lowest aerosol number concentrations demonstrates the importance of understanding and representing aerosol sources and aerosol recycling in the central Arctic. The result also highlights that long-range advection plays an important role in supplying low-level Arctic MPS clouds with moisture and CCN, as the relatively high concentrations of moisture and aerosols during the case study were associated with southerly/southeasterly winds. Previous observations (Bigg et al., 1996, 2001; Leck and Persson, 1996, Lundén et al., 2007; Chang et al., 2011; Heintzenberg et al., 2015; Shupe et al., 2022; Dada et al., 2022) have shown that remote sources can contribute to the aerosol number concentrations over the central Arctic Ocean. To address all these questions, more simultaneous aerosol and cloud observations are needed in the central Arctic during all seasons, especially in terms of their vertical distribution. Furthermore, while the lower cloud layer was relatively insensitive to varying aerosol concentrations, it showed stronger sensitivity to the large-scale wind speed. This underlines the importance of considering Arctic clouds and cloud microphysical properties in a meteorological context.





**Appendix**

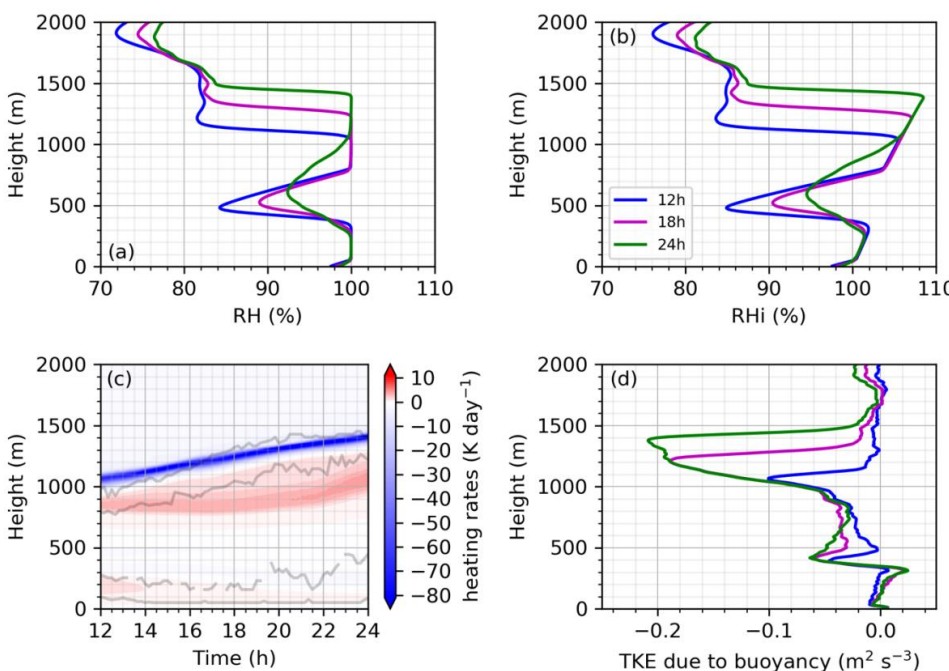

**Figure A1.** Simulated vertical profiles of (a) relative humidity, (b) relative humidity with respect to ice. (c) Model-domain average simulated vertical distribution of heating rates and (d) simulated vertical profiles of buoyancy induced turbulent kinetic energy (TKE). Note that simulated profiles are horizontal averages over the model domain.

| | case study (dN, Dp, σ) | | ice drift (dN, Dp, σ) | |
|---|---|---|---|---|
| | Aitken mode | accumulation mode | Aitken mode | accumulation mode |
| 25th percentile | 18, 36, 1.27 | 25, 175, 1.26 | 3,28,1.29 | 2,144,1.47 |
| 50th percentile | 29,36,1.30 | 34,169,1.31 | 8,28,1.38 | 6,129,1.52 |
| 75th percentile | 38,35,1.34 | 46,158,1.35 | 19,26,1.46 | 12,114,1.72 |

**Table A1.** Lognormal distribution parameters (integrated particle concentration dN, modal diameter Dp and standard deviation σ) of the particle size distribution, calculated for the simulated case study (12:00 to 24:00 UTC on 18 August) and the whole ice drift period (14 August - 14 September).

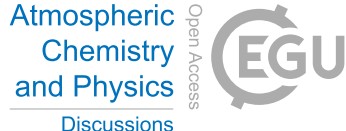

*Data availability.* Modeling data sets used in this study are available at https://bolin.su.se/data/bulatovic-2022-cloud-structure-1, last access: 1 December 2022 (Bulatovic et al., 2022). The observational data can also be accessed through the Bolin Centre Database (Leck et al., 2020).

*Author contribution:* IB, AE and CL developed the idea for the study. The model simulations were further discussed with MT and other members of the AO2018 expedition. The model initial forcing was setup by IB with help from JS. IB performed the model simulations and data analysis. All co-authors provided important comments on data interpretation. IB wrote the main part of the paper with input from all authors and specific input by MT on Sect. 3.1. CL provided Figure 5 in the manuscript.

*Competing interests.* The authors declare that they have no conflict of interest.

*Acknowledgments.* We thank Linn Karlsson and Paul Zieger for sharing their DMPS measurements with us and for helping us choose the period that was investigated as a case study. We thank Mike Lawler for providing us with the kappa values of the Aitken and accumulation mode particles, which we used in our simulations. We also thank Grace Porter for helping us choose the default value for the ice crystal number concentration. The Swedish Polar Research Secretariat (SPRS) provided access to the I/B Oden and logistical support in collaboration with the U.S. National Science Foundation. We are grateful to Co-Chief Scientists Patricia Matrai and co-author Caroline Leck for planning, technical support, and coordination of AO2018, to the SPRS logistical staff and to I/B Oden's Captain Mattias Peterson and his crew for expert field support.
The computations and data handling were enabled by resources provided by the Swedish National Infrastructure for Computing (SNIC) at the National Supercomputer Centre (NSC) partially funded by the Swedish Research Council through grant agreement no. 2018-05973.

*Financial support.* This research has been supported by the Swedish Science Foundation (Vetenskapsrådet; grant no. 2015-05318 and no. 2016-03518), the Bolin Centre for Climate Research (RA2), the Knut- and Alice-Wallenberg Foundation within the ACAS project (Arctic Climate Across Scales, project no. 2016.0024)) and the European Union's Horizon 2020 research and innovation programme (FORCeS, grant no. 821205 and CRiceS, grant no. 101003826).

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
