# Peer review of "Large-eddy simulation of a two-layer boundary-layer cloud system from the Arctic Ocean 2018 expedition"

_Atmospheric Chemistry and Physics, 2022_

## Referee Comment (RC1)

This paper uses Large Eddy Simulation (LES) modelling to investigate the effects of aerosol and ice concentrations on a 2-layer Arctic stratocumulus cloud case study. The control case produces a cloud that seems to match the observations very well (although a more detailed comparison to observations could be performed; see below). The effects of changing the aerosol concentration is investigated with the model and shows an increase in LWP with increasing aerosols. At low aerosol concentration the upper cloud layer dissipates with large shortwave and longwave surface radiative impacts (a change from an overall net SW+LW warming of the surface in the control case to a near-zero net radiative surface flux; this will likely affect surface melting). The effects of ice concentrations and divergence are also investigated but they only have small impacts. Increasing the wind speed causes the lower cloud layer to dissipate while the upper one remains (but gets thinner). The removal of the lower layer causes an increase in the longwave surface cooling and reduces the net surface LW+SW flux by around 50%.

Overall, the study is well designed and the paper is well written with some interesting results. The model seems to produce realistic results for the control case, although the available observations could be used to test the model more thoroughly (e.g., a comparison of radar reflectivity). And there are some questions about how realistic the treatment of entrainment is (particularly its variation with aerosol concentrations). I also think that the paper could do a better job of describing some of the shortwave vs longwave effects of the upper cloud layer vs that of the lower cloud layer. Please see the comments below for more details on all of these issues.

I recommend the publication of this paper once the points below are addressed.

**Main issues**

Section 2.2 – the model details give no information about whether the model includes droplet sedimentation. It would be good to provide this information and to comment on what that implies for entrainment changes as a function of aerosol changes. More discussion on the literature of entrainment effects would also be good.

Fig. 2 – is it possible to make versions of this from the modelled data for comparison to the observations? E.g., a comparison of the radar reflectivity would help to test whether the model precipitation rates and snow/graupel amounts are accurate since this is likely to be an important process that determines the sensitivity to aerosols. Or if not, it could be mentioned as something that could be done in the future.

L432 – could the smaller IWP in the run with the larger LWP be due to the fact that there was lots of graupel in that run (Fig. 8i) so that graupel water path would actually be quite large (maybe around 5 g/m2). So if you combined the ice, snow and graupel to give a total ice water path it may be larger than the other runs? Does the IWP from the observations also include snow and graupel?

L444 – It would be worth mentioning the SW+LW effect – i.e. that very low aerosol can prevent the net surface energy input to the surface (that could cause melting).

L576 – "This means that the lower layer became partially transparent to radiation so that the longwave radiative effect dominated instead of the shortwave effect, i.e., a lower aerosol concentration resulted in a cooling of the surface." – this is a little bit imprecise and unclear. How

about :- "This meant that the both cloud layers were partially transparent to shortwave radiation, which increased the shortwave heating of the surface. However, both cloud layers also became too thin to emit significant longwave radiation which increased the longwave cooling of the surface. The longwave radiative cooling effect dominated over the shortwave warming effect so that the lower aerosol concentration resulted in a cooling of the surface."

- Although it's not clear from the rest of the paper whether you determined whether it was the disappearance of the top cloud layer in aero\_id\_low that led to the increase in net surface shortwave, or whether it was the thinning of the lower layer. And similarly for the longwave effects the lower cloud layer still looks to be there in aero\_id\_low. Fig. 14 suggests that the removal of just the lower cloud layer in the wind\_8.5 experiment leads to an increase in the LW surface cooling from -4 to -12 W/m2 by the end of the simulation (a difference of -8 W/m2), whereas Fig. 10 shows a reduction from -4 to -40 W/m2 due to the removal of the upper cloud layer (with the lower cloud layer still present, although thinned out somewhat Fig. 9). This suggests that it is the removal of the upper cloud layer in aero\_id\_low that is having the bigger impact on the longwave and shortwave fluxes?
- This should also be addressed in the abstract. Currently you write :-
  - • "The investigated cloud structure is persistent unless there are low aerosol number concentrations (≤ 5 cm-3), which cause the upper cloud layer to dissipate, or high large-scale wind speeds (~ 8.5 m s-1), which erode the lower inversion and the related cloud layer. These types of changes in cloud structure lead to a substantial reduction of the net longwave radiation at the surface due to a lower emissivity or higher altitude of the remaining cloud layer."
- However, you should also mention the importance of the increased surface warming from the shortwave and increased LW cooling when the upper layer is eroded. It would also be good to talk about the net SW+LW effect in the abstract since this will help determine surface melting. I.e., the very low aerosol case leads to a near-zero net radiative heating at the surface, which may reduce or prevent surface melting. Although the wind effect on the surface net warming (lower layer only) is smaller.

L584 – It would be good to document what happens to the total ice water path here as well (ice+snow+graupel). Also, it is interesting that IWP increases towards the end of the simulation in the lower ice concentration cases, so that it matches the higher ice concentration cases – can you say something on why this is?

L593 – "A noticeable effect on the longwave radiation was only obtained in the experiments where the lower cloud layer became optically thin or completely dissipated (i.e., the simulations with the lowest aerosol number concentrations and the highest wind speed, respectively)."

- I'm not sure that I agree with this since the removal of the upper cloud seemed to have an even larger large longwave effect – see above.

**Figures**

Fig. 2 – It would be good to have some titles and colorbar labels on this figure.

Fig. 5 – It's not quite clear what dN refers to and why the x-axis is the modal diameter. Is this instead showing dN/dlogDp with Dp being just the aerosol diameter? I.e., does the integral under the curves give the total number?

Fig. 6 – the colours for 12 and 18h are not very colorblind friendly – I'm finding it hard to distinguish them.

**Typos etc.**

L23 – "capped by a lower temperature inversion" – do you mean a smaller magnitude inversion or a lower-altitude one?

L24 – "The investigated cloud structure" – better as "The simulated cloud structure" to show that this was the result of modelling rather than from observations.

L27 – "net longwave radiation at the surface" – would be good to say that this is the "net downwelling longwave radiation" for clarity.

L85 – "difficulty to simulate" -> "difficulty simulating".

L150 – "minimize the risk of sampling pollution from the ship, I/B Oden was turned approximately upwind" – it would be good to mention that the ship exhausts are (presumably) at the rear of the ship relative to the instruments.

L359 – "likely due to that large-scale advection is not explicitly considered in the LES" -> "likely due to the fact that large-scale advection is not explicitly considered in the LES"

L419 - "less (more) aerosols" -> "fewer (more) aerosols"

L568 – "solar part of the spectrum" should be "shortwave part of the spectrum" since the solar spectrum covers the whole range (although peaking in SW of course).

L570 – "for 4W/m2" -> "by 4 W/m2".

L572 – "When Aitken and accumulation mode aerosol number concentrations in. being representative of the whole ice drift period" -> "When Aitken and accumulation mode aerosol number concentrations that were representative of the whole ice drift period"

L573 – "then the total LWP decreased substantially (up to 150 g m-2)" -> "then the total LWP decreased substantially (by up to 150 g m-2)"

L575 - "representative of the lowest observed percentile (=5 cm-3)," -> "representative of the 25th percentile of the ice drift observations (=5 cm-3),"

---

## Author Comment (AC1)

Reply to referee's #1 comments on manuscript:

This paper uses Large Eddy Simulation (LES) modelling to investigate the effects of aerosol and ice concentrations on a 2-layer Arctic stratocumulus cloud case study. The control case produces a cloud that seems to match the observations very well (although a more detailed comparison to observations could be performed; see below). The effects of changing the aerosol concentration is investigated with the model and shows an increase in LWP with increasing aerosols. At low aerosol concentration the upper cloud layer dissipates with large shortwave and longwave surface radiative impacts (a change from an overall net SW+LW warming of the surface in the control case to a near-zero net radiative surface flux; this will likely affect surface melting). The effects of ice concentrations and divergence are also investigated but they only have small impacts. Increasing the wind speed causes the lower cloud layer to dissipate while the upper one remains (but gets thinner). The removal of the lower layer causes an increase in the longwave surface cooling and reduces the net surface LW+SW flux by around 50%.

Overall, the study is well designed and the paper is well written with some interesting results. The model seems to produce realistic results for the control case, although the available observations could be used to test the model more thoroughly (e.g., a comparison of radar reflectivity). And there are some questions about how realistic the treatment of entrainment is (particularly its variation with aerosol concentrations). I also think that the paper could do a better job of describing some of the shortwave vs longwave effects of the upper cloud layer vs that of the lower cloud layer. Please see the comments below for more details on all of these issues.

I recommend the publication of this paper once the points below are addressed.

We thank the reviewer for carefully reading the manuscript and for the constructive comments.

**Main issues**

Section 2.2 – the model details give no information about whether the model includes droplet sedimentation. It would be good to provide this information and to comment on what that implies for entrainment changes as a function of aerosol changes. More discussion on the literature of entrainment effects would also be good.

We agree with the reviewer that sedimentation can influence entrainment rates and their dependency on aerosol changes. However, we would argue that those changes should be more pronounced in the cases where the free troposphere (FT) is drier than the boundary layer (BL) than in the cases where the FT represents a moisture source for the BL, as is the case here. Nevertheless, in the Sect. 2.2., we have now added: "Sedimentation is not included in the model. Its introduction could lead to additional changes in simulated cloud properties (cf. Ackerman et al., 2004; Hoffmann and Feingold, 2019)."

Fig. 2 – is it possible to make versions of this from the modelled data for comparison to the observations? E.g., a comparison of the radar reflectivity would help to test whether the model precipitation rates and snow/graupel amounts are accurate since this is likely to be an important process that determines the sensitivity to aerosols. Or if not, it could be mentioned as something that could be done in the future.

We agree that this type of model output would be useful. However, it is not available in the current MIMICA version. It would require substantial work to either get the right output to run an offline radar simulator or implement a radar reflectivity parameterization in the model. Moreover, Figure 2 is included more to describe the particulars of the simulated situation than to provide a detailed evaluation. We believe that performing a detailed evaluation of the LES is beyond the scope of this paper. We have added a note on this issue in the manuscript: "To further analyze precipitation rates and specifically the amount of solid precipitation, comparing observed and modelled radar reflectivity would be a useful approach. However, calculation of radar reflectivity based on the model output is beyond the scope of this paper."

L432 – could the smaller IWP in the run with the larger LWP be due to the fact that there was lots of graupel in that run (Fig. 8i) so that graupel water path would actually be quite large (maybe around 5 g/m2). So if you

combined the ice, snow and graupel to give a total ice water path it may be larger than the other runs? Does the IWP from the observations also include snow and graupel?

The simulated IWP shown in Fig. 10b is the total IWP. Since the collection of cloud droplets by ice becomes less efficient in experiments with more LWP (i.e. when the cloud droplets are smaller in size), this process should be the main reason for the IWP decrease simulated in the experiments with a higher aerosol number concentration. We have now added to the Fig. 7 caption that both LWP and IWP are total values.

L444 – It would be worth mentioning the SW+LW effect – i.e. that very low aerosol can prevent the net surface energy input to the surface (that could cause melting).

We agree with the reviewer on this point. We now mention the net SW+LW effect with different sensitivity tests in all sections.

L576 – "This means that the lower layer became partially transparent to radiation so that the longwave radiative effect dominated instead of the shortwave effect, i.e., a lower aerosol concentration resulted in a cooling of the surface." – this is a little bit imprecise and unclear. How about :- "This meant that the both cloud layers were partially transparent to shortwave radiation, which increased the shortwave heating of the surface. However, both cloud layers also became too thin to emit significant longwave radiation which increased the longwave cooling of the surface. The longwave radiative cooling effect dominated over the shortwave warming effect so that the lower aerosol concentration resulted in a cooling of the surface."

We agree with the reviewer that our previous statement was not clear enough. We have now changed it to: "Without the upper cloud, and with a partially transparent lower cloud, the shortwave radiative flux at the surface increased. However, the optically thin lower cloud layer also emitted less longwave radiation, which led to increased longwave cooling of the surface. These two radiative fluxes were approximately balanced so the net radiative effect in this sensitivity test was around zero, in contrast to all other aerosol sensitivity simulations where the net radiative effect of the cloud structure was positive."

- Although it's not clear from the rest of the paper whether you determined whether it was the disappearance of the top cloud layer in aero_id_low that led to the increase in net surface shortwave, or whether it was the thinning of the lower layer. And similarly for the longwave effects – the lower cloud layer still looks to be there in aero_id_low. Fig. 14 suggests that the removal of just the lower cloud layer in the wind_8.5 experiment leads to an increase in the LW surface cooling from -4 to -12 W/m2 by the end of the simulation (a difference of -8 W/m2), whereas Fig. 10 shows a reduction from -4 to -40 W/m2 due to the removal of the upper cloud layer (with the lower cloud layer still present, although thinned out somewhat – Fig. 9). This suggests that it is the removal of the upper cloud layer in aero_id_low that is having the bigger impact on the longwave and shortwave fluxes?
- This should also be addressed in the abstract. Currently you write :-
   o "The investigated cloud structure is persistent unless there are low aerosol number concentrations (≤ 5 cm-3), which cause the upper cloud layer to dissipate, or high large-scale wind speeds (~ 8.5 m s-1), which erode the lower inversion and the related cloud layer. These types of changes in cloud structure lead to a substantial reduction of the net longwave radiation at the surface due to a lower emissivity or higher altitude of the remaining cloud layer."
- However, you should also mention the importance of the increased surface warming from the shortwave and increased LW cooling when the upper layer is eroded. It would also be good to talk about the net SW+LW effect in the abstract since this will help determine surface melting. I.e., the very low aerosol case leads to a near-zero net radiative heating at the surface, which may reduce or prevent surface melting. Although the wind effect on the surface net warming (lower layer only) is smaller.

We agree with this point as well. In the abstract, we have now written: "The changes in cloud structure alter both the short- and longwave cloud radiative effect at the surface. This results in changes in the net radiative effect of the modelled cloud system, which can impact the surface melting/freezing." We have tried to mention everything relevant but to keep it general.

L584 – It would be good to document what happens to the total ice water path here as well (ice+snow+graupel). Also, it is interesting that IWP increases towards the end of the simulation in the lower ice concentration cases, so that it matches the higher ice concentration cases – can you say something on why this is?

Figure 12b shows the total IWP, which increases with time in all the ICNC experiments (also note the logarithmic scale; the absolute value increases approximately at the same rate in all three simulations). Moreover, in almost all simulations in this study, the IWP suddenly increases around 17h. This most probably corresponds to when the cloud becomes cold enough; so, the ice formation becomes efficient.

We agree that this was not explained in detail. Therefore, we have now written: "Moreover, in the control simulation, the IWP suddenly increases ~16-17h. This is most probably due to the fact that the cloud becomes cold enough to trigger efficient ice formation. Both LWP and IWP then continues to increase for ~4h, indicating a continuous cloud water production, until LWP finally starts to decrease around 21h. Note, however, that, first, the increasing LWP is much larger than the increase in IWP and, second, the decrease in LWP after 21h is larger than the increase in IWP by an order of magnitude. This is also in agreement with previous findings (cf. Dimitrelos et al., 2020)."
However, we have added this to the Sect. 4.2.1. since this sudden IWP increase with time can already be seen in the experiments with different aerosol number concentrations.

L593 – "A noticeable effect on the longwave radiation was only obtained in the experiments where the lower cloud layer became optically thin or completely dissipated (i.e., the simulations with the lowest aerosol number concentrations and the highest wind speed, respectively)."
      - I'm not sure that I agree with this since the removal of the upper cloud seemed to have an even larger large longwave effect – see above.

This statement is strictly correct since the net longwave radiative flux at the surface does not change significantly in any experiment except in the two tests we have mentioned. As long as the lower cloud layer is optically thick, the upper cloud layer does not significantly impact the longwave flux at the surface since the lower cloud layer is closer to the surface and warmer. But as soon as the lower cloud layer becomes optically thin, the influence of the upper cloud layer is also important (which can be seen when comparing the aero_id_low and wind_8.5 simulations). It is important to note here that the lower cloud layer doesn't have to be very optically thick; the upper layer can still dominate the total LWP. It just has to be thick enough to be a black body; then, the surface longwave net radiation will be determined by the surface temperature and the temperature of the lower cloud layer.

In the Conclusions section, we have now added one more sentence to clarify this: "However, when the lower cloud layer was not optically thick, the upper cloud layer also affected the surface longwave radiation (the surface cooling was much stronger in the simulation with the lowest aerosol concentration than in the simulation with the highest wind spend due to the lack of an upper cloud layer in the first experiment)."

**Figures**

Fig. 2 – It would be good to have some titles and colorbar labels on this figure.

The colourbar units have now been added.

Fig. 5 – It's not quite clear what dN refers to and why the x-axis is the modal diameter. Is this instead showing dN/dlogDp with Dp being just the aerosol diameter? I.e., does the integral under the curves give the total number?

The y-axis shows dN (dN/dlogDp multiplied with dlogDp). The modal integrals under the fitted curve correspond to dN values, which are the numbers used as the model input parameters (also shown in Table A1). Using dN/dlogDp would not correspond to the numbers we used in our simulations. We understand this was not explained well in the previous manuscript version. Thus, we have added: "Note that the dN values shown in Table A1 represent the modal integrals under the fitted curve in Fig. 5 (y-axis; dN/dlogDp multiplied with dlogDp) and are the numbers used as the model input parameters."
The x-axis label is now changed to aerosol diameter.

Fig. 6 – the colours for 12 and 18h are not very colorblind friendly – I'm finding it hard to distinguish them.

This is now updated. Also, the colours in Fig. A1 are changed accordingly.

**Typos etc.**

L23 – "capped by a lower temperature inversion" – do you mean a smaller magnitude inversion or a lower-altitude one?

We have now changed it to a "lower-altitude" inversion.

L24 – "The investigated cloud structure" – better as "The simulated cloud structure" to show that this was the result of modelling rather than from observations.

Ok, we have changed it.

L27 – "net longwave radiation at the surface" – would be good to say that this is the "net downwelling longwave radiation" for clarity.

This sentence is not the same as it was in the first version of the manuscript.

L85 – "difficulty to simulate" -> "difficulty simulating".

Ok.

L150 – "minimize the risk of sampling pollution from the ship, I/B Oden was turned approximately upwind" – it would be good to mention that the ship exhausts are (presumably) at the rear of the ship relative to the instruments.

This has now been added.

L359 – "likely due to that large-scale advection is not explicitly considered in the LES" -> "likely due to the fact that large-scale advection is not explicitly considered in the LES"

This is added.

L419 – "less (more) aerosols" -> "fewer (more) aerosols"

This is changed.

L568 – "solar part of the spectrum" should be "shortwave part of the spectrum" since the solar spectrum covers the whole range (although peaking in SW of course).

We agree, this has been changed.

L570 – "for 4W/m2" -> "by 4 W/m2".

It is changed.

L572 – "When Aitken and accumulation mode aerosol number concentrations in. being representative of the whole ice drift period" -> "When Aitken and accumulation mode aerosol number concentrations that were representative of the whole ice drift period"

We have changed this.

L573 – "then the total LWP decreased substantially (up to 150 g m-2)" -> "then the total LWP decreased substantially (by up to 150 g m-2)"

We have added "by".

L575 – "representative of the lowest observed percentile (=5 cm-3)," -> "representative of the 25th percentile of the ice drift observations (=5 cm-3),"

This is also changed.

---

## Author Comment (AC2)

Reply to referee's #2 comments on manuscript:

The paper examines summer-time Arctic boundary layer clouds, observed during a shipborne campaign. For a few hours clouds showed a two-layer vertical structure. The paper uses LES to reproduce the observations and to also perform a sensitivity analysis. The authors find that decreased aerosol concentrations and greater windspeed would erode one or the other layer.

The paper is well written and the figures are of good quality. I have one major and a few minor recommendations that the authors should resolve before publication.

We thank the reviewer for carefully reading the manuscript and the constructive comments.

Major comment

Section 4.2.3: Given the great sensitivity to windspeed, the topic appears under-explored. Perhaps the authors could provide additional information that helps to understand how erosion of the lower cloud layer was facilitated:
How much of increase in surface turbulent heat fluxes was found for greater windspeed?
How did the TKE profiles change across simulations?
What explains the increase in column total water?

We agree with the reviewer that this topic could have been explained in more detail.
In the simulated case, there is generally a small surface heat flux because the boundary layer (BL) is in principle well-mixed (i.e. near-neutral temperature profile). With weaker (stronger) wind speeds, there is less (more) mixing but it doesn't alter the surface turbulent fluxes much as long as the BL is already well-mixed (this has also been checked and confirmed by the model output). The changes in the wind speed modulate the mechanical mixing; it is larger with higher wind speed and vice versa. More substantial mixing in the experiment with higher wind speeds thus adds additional mechanical turbulence to the buoyancy turbulence. It leads to erosion of the lower part of the BL, coupling the upper cloud layer with the surface and inducing more entrainment. These changes also affect the LWP.

In the Sect. 4.2.3, it has now been added: "With stronger winds (i.e., wind_8.5), there is more mechanical mixing from the surface, which erodes the lower-altitude temperature inversion (not shown) and causes the lower cloud layer to dissipate after ~4 hours of simulation (Fig. 15c). This further connects the whole BL layer and induces more entrainment over the top of the upper cloud layer. The net effect is a single-layer cloud structure in a slightly deeper BL with a somewhat higher cloud top. This change also affects the LWP, which is reduced both by the dissipation of the lower cloud layer and by the somewhat lower cloud water concentrations in the upper cloud layer; this also reduces the IWP (see Fig. 14a and 14b). Interestingly, this difference to the control run is present immediately after the spin-up, when only a weak moisture inversion is present; this is then enhanced as the BL deepens and the moisture inversion strengthens. Presumably, the entrainment of warmer air accumulated over time has a larger effect decreasing the cloud liquid water than entraining of absolutely moister air across the cloud top. With the lower wind speeds (i.e., wind_3.5 simulation), the opposite happens. There is less mechanical mixing and therefore a shallower surface-based BL develops. In this shallow BL, a lower cloud layer forms capped by a lower-altitude inversion, which decouples the upper cloud layer from the surface."

We have decided not to add additional figures since the manuscript is already figure-rich.

Minor comments

Fig. 1: For geographical context either Latitude lines or labels of nearby landmasses would be helpful.

We have added labels of nearby landmasses.

Fig. 5: Perhaps changed y label to dN/dlogDp.

The modal integrals under the fitted curve correspond to dN values, which are also the numbers used as the model input parameters (shown in Table A1). Using dN/dlogDp would not correspond to the numbers we used in our simulations. We understand this was not explained well in the previous manuscript version. Thus, we have added: "Note that the dN values shown in Table A1 represent the modal integrals under the fitted curve in Fig. 5 (y-axis; dN/dlogDp multiplied with dlogDp) and are the numbers used as the model input parameters."

Section 2.2: A few details should be added here:

How was temperature and moisture advection handled?

In the Model section, it has now been added: "The scalar advection follows a Lex-Wendroff flux limited method described in Durran (2010) with periodic boundary conditions but horizontal large-scale advection tendencies are set to zero.".

Was aerosol treated diagnostically or prognostically?

In the previous version of the manuscript, it was written: "The model includes a two-moment aerosol module with a prescribed number of lognormal aerosol modes (Ekman et al., 2006).". We have modified this: "The model includes an interactive two-moment aerosol module where an arbitrary number of lognormal aerosol modes can be defined (Ekman et al., 2006). Aerosol particles that are activated into cloud droplets can be scavenged from the model domain through precipitation. The aerosol mass within hydrometeors is tracked by the model and is released back to the atmosphere when hydrometeors evaporate or sublimate.".

Does the simulation consider the water vapor and ozone column above the model domain?

Yes, we have used a standard polar atmosphere profile above the top of the model domain, including both water vapour and ozone. This information has now been added to the Sect. 2.2.

Was the diurnal cycle of solar insolation taken into account?

Yes, this information is now also added to the Model section.

ll. 308-309: Please explain which instrument or technique was involved in estimating hygroscopicities.

In the new version of the manuscript, we have added: "The estimated kappa values were based on *in situ* observations of aerosol particle chemical composition, using a thermal desorption chemical ionization mass spectrometer (TDCIMS), for the period of interest. The calculation used a composition-weighted average of assumed kappa values for the various constituents determined (e.g., sulfuric acid, assumed kappa=1.19; ammonium bisulfate, assumed kappa =0.8)."

---

## Referee Report (RR1)

**Review of revised version of Bulatovic et al. following first review**

The authors have done a good job of addressing the comments from the first review. I have just a few small further suggestions. The line numbers are from the original ACPD manuscript :-

**Original comment :-**

 L432 – could the smaller IWP in the run with the larger LWP be due to the fact that there was lots of graupel in that run (Fig. 8i) so that graupel water path would actually be quite large (maybe around 5 g/m2). So if you combined the ice, snow and graupel to give a total ice water path it may be larger than the other runs? Does the IWP from the observations also include snow and graupel?

**Author Response :-**

The simulated IWP shown in Fig. 10b is the total IWP. Since the collection of cloud droplets by ice becomes less efficient in experiments with more LWP (i.e. when the cloud droplets are smaller in size), this process should be the main reason for the IWP decrease simulated in the experiments with a higher aerosol number concentration. We have now added to the Fig. 7 caption that both LWP and IWP are total values.

**Additional comments :-**

OK, but the text ("The reason is that the collection of cloud droplets by ice becomes more efficient in experiments with less LWP (i.e., when the droplets are larger; not shown). ") is a little confusing as it doesn't explain that when the LWP is lower we have fewer droplets, which are therefore larger (despite the lower LWP). I suggest "The reason is that in the experiments with less LWP there are also fewer droplets, so that the droplets are larger (the reduction in number dominates over the reduction in LWP) and therefore the collection of cloud droplets by ice becomes more efficient (not shown)."

Also, total IWP and LWP is not particularly clear - it would be good to specify that you mean ice+snow+graupel and liquid+rain (assuming that this is what you mean by total LWP).

**Original comment :-**

Fig. 5 – It's not quite clear what dN refers to and why the x-axis is the modal diameter. Is this instead showing dN/dlogDp with Dp being just the aerosol diameter? I.e., does the integral under the curves give the total number?

**Author Response :-**

The y-axis shows dN (dN/dlogDp multiplied with dlogDp). The modal integrals under the fitted curve correspond to dN values, which are the numbers used as the model input parameters (also shown in Table A1). Using dN/dlogDp would not correspond to the numbers we used in our simulations. We understand this was not explained well in the previous manuscript version. Thus, we have added: "Note that the dN values shown in Table A1 represent the modal integrals under the fitted curve in Fig. 5 (y-axis; dN/dlogDp multiplied with dlogDp) and are the numbers used as the model input parameters."

**Additional comments :-**

Ok, but perhaps the phrase "integrated particle concentration dN" in the caption would be better as "particle concentration in each size bin, dN".

"Note that the dN values shown in Table A1 represent the modal integrals under the fitted curve in Fig. 5 (y-axis; dN/dlogDp multiplied with dlogDp)" seems to contradict what you just said. It would be better to call the values in Table A1 "N" rather than "dN" and to say that they "represent the total number of aerosols in each mode from the fitted curves in Fig. 5 (sum of dN values)…"

---

## Author Response (AR2)

Reply to referee's #1 comments on manuscript:

We thank the reviewer for carefully reading the manuscript and for additional constructive comments.

OK, but the text ("The reason is that the collection of cloud droplets by ice becomes more efficient in experiments with less LWP (i.e., when the droplets are larger; not shown). ") is a little confusing as it doesn't explain that when the LWP is lower we have fewer droplets, which are therefore larger (despite the lower LWP). I suggest "The reason is that in the experiments with less LWP there are also fewer droplets, so that the droplets are larger (the reduction in number dominates over the reduction in LWP) and therefore the collection of cloud droplets by ice becomes more efficient (not shown)."

We agree with the reviewer that the previous statement could have been clearer. We have now changed it to: "The reason is that in the experiments with less LWP there are fewer droplets, so that the droplets are larger (the reduction in number dominates over the reduction in LWP) and therefore the collection of cloud droplets by ice becomes more efficient (not shown)."

Also, total IWP and LWP is not particularly clear - it would be good to specify that you mean ice+snow+graupel and liquid+rain (assuming that this is what you mean by total LWP).

We have now added the clarification to the Fig. 7 caption.

Ok, but perhaps the phrase "integrated particle concentration dN" in the caption would be better as "particle concentration in each size bin, dN".

We have now changed it to "particle concentration in each size bin, dN".

"Note that the dN values shown in Table A1 represent the modal integrals under the fitted curve in Fig. 5 (yaxis; dN/dlogDp multiplied with dlogDp)" seems to contradict what you just said. It would be better to call the values in Table A1 "N" rather than "dN" and to say that they "represent the total number of aerosols in each mode from the fitted curves in Fig. 5 (sum of dN values)..."

The A1 caption is now written as: "**Table A1.** Distribution parameters ("total particle concentration in each mode, N; modal diameter, Dp and standard deviation $\sigma$) of the particle size distribution, calculated for the simulated case study (12:00 to 24:00 UTC on 18 August) and the whole ice drift period (14 August - 14 September). Note that N values shown in Table A1 represent the total number of aerosols in each mode from the fitted curves in Fig. 5 (sum of dN values) and are the numbers used as the model input parameters."

Reply to referee's #2 comments on manuscript:

The revisions improved the article substantially. I only have one minor comment and recommend publication after resolving it.

We thank the reviewer for carefully reading the manuscript and for additional constructive comment.

Fig. 5: In their response, the authors express that indeed the integral form, "dN", is shown, because the model input is structured that way. However, in line 310 the model is given "log-normal modes". This contradiction should be resolved. Also, if "dN" (instead of "dN/dlogD") is used, wouldn't it be necessary to provide bin information for others to compare?

We have now removed "log-normal" and instead written: "The aerosol number size distribution was represented by two modes (accumulation and Aitken) that were fitted to the observed values (see Fig. 5 and Table A1)." We refer to the A1 table so that readers can check the model input parameters. The A1 caption has also been slightly modified for clarification sake.